# Analysis of sub-kilobase chromatin topology reveals nano-scale regulatory interactions with variable dependence on cohesin and CTCF

Abrar Aljahani[1], Peng Hua [2,3], Magdalena A. Karpinska [1], Kimberly Quililan[1], James O. J. Davies [2✉] & A. Marieke Oudelaar [1✉]

Enhancers and promoters predominantly interact within large-scale topologically associating domains (TADs), which are formed by loop extrusion mediated by cohesin and CTCF. However, it is unclear whether complex chromatin structures exist at sub-kilobase-scale and to what extent fine-scale regulatory interactions depend on loop extrusion. To address these questions, we present an MNase-based chromosome conformation capture (3C) approach, which has enabled us to generate the most detailed local interaction data to date (20 bp resolution) and precisely investigate the effects of cohesin and CTCF depletion on chromatin architecture. Our data reveal that *cis*-regulatory elements have distinct internal nano-scale structures, within which local insulation is dependent on CTCF, but which are independent of cohesin. In contrast, we find that depletion of cohesin causes a subtle reduction in longer-range enhancer-promoter interactions and that CTCF depletion can cause rewiring of regulatory contacts. Together, our data show that loop extrusion is not essential for enhancer-promoter interactions, but contributes to their robustness and specificity and to precise regulation of gene expression.

[1] Max Planck Institute for Multidisciplinary Sciences, Göttingen, Germany. [2] MRC Molecular Haematology Unit, MRC Weatherall Institute of Molecular Medicine, University of Oxford, Oxford, UK. [3]Present address: State Key Laboratory of Reproductive Medicine, Nanjing Medical University, Nanjing, China. ✉email: james.davies@imm.ox.ac.uk; marieke.oudelaar@mpinat.mpg.de

Mammalian gene expression patterns are controlled by enhancers, which form specific interactions with the promoters of their target genes to transfer activating signals[1]. Since these elements can be separated by large genomic distances, the specificity of enhancer–promoter interactions is dependent on the 3D organization of the genome in the nucleus[2]. Mammalian chromosomes are organized into compartments and topologically associating domains (TADs)[3–6]. Compartments reflect the separation of euchromatin and heterochromatin[3], whereas TADs are formed by an active process of loop extrusion halted by boundary elements[7,8]. Accumulating evidence has identified cohesin as the key loop extruding factor and CTCF-binding elements as the major boundaries in mammalian cells[9].

As interactions between enhancers and promoters predominantly occur within TADs[10], these domains are thought to be relevant for the regulation of the specificity of enhancer–promoter communication. Previous studies have shown that perturbations of CTCF-binding elements at TAD boundaries result in changes in chromatin structure, which can lead to rewiring of enhancer–promoter interactions and ectopic activation of genes by enhancers that are normally located in a neighboring TAD[8,11–19]. However, more recent genetic dissections of TAD boundaries have demonstrated that the impact of TAD rearrangements on gene expression is context-dependent and sometimes very mild, thus questioning the importance of TADs for gene regulation[20–23]. Due to potential redundancy between CTCF-binding elements, it is challenging to deduce the general function of CTCF and TADs in the regulation of enhancer–promoter specificity based on perturbations of individual TAD boundaries.

The development of acute protein degradation approaches has made it possible to investigate the impact of genome-wide depletions of key architectural proteins on genome organization and gene expression[24]. Using such approaches, it has been shown that depletion of cohesin, its loading factor NIPBL, and CTCF result in large-scale changes in the 3D organization of the genome and a loss of TADs[25–31]. However, surprisingly, the effects of these depletions on gene expression are relatively mild. It has been suggested that loop extrusion is only required for mediating very long-range enhancer–promoter interactions, which could explain why only a small subset of genes is misregulated upon the depletion of cohesin and CTCF[32,33]. Furthermore, it has recently been proposed that the relationship between enhancer–promoter interaction strength and gene activity is not linear[34,35]. Another possible explanation is that loop extrusion is particularly important for regulatory interactions in the context of dynamic gene expression changes, as has been suggested based on studies of the functions of cohesin and CTCF during neuronal differentiation and upon inflammatory signaling[31,36,37].

A limitation of the current studies is that the effects of depletion of cohesin and CTCF on chromatin architecture have been analyzed with genome-wide Chromosome Conformation Capture (3C)[25–30,32] and imaging approaches[29,38–40] with relatively low resolution. Although these studies have given important insights into the roles of cohesin and CTCF in large-scale genome organization, the functions of these proteins in regulating chromatin interactions at a smaller scale remain unresolved. To address this, it is important to analyze the effects of cohesin and CTCF depletion on chromatin architecture with approaches that enable analysis at very high resolution and sensitivity and which can detect potentially subtle changes in fine-scale chromatin structures, including the specific regulatory interactions between enhancers and promoters which control gene expression. This would contribute to a better understanding of the roles of loop extrusion and TADs in gene regulation and the relationship between genome structure and function.

Conventional 3C methods[4,41] are limited in their resolution by sequencing depth, library complexity, and the distribution of the recognition sites of the restriction enzymes used for chromatin digestion[42]. The latter can be overcome by the use of micrococcal nuclease (MNase), which digests the genome largely independent of DNA sequence. MNase digestion was initially implemented in the Micro-C approach in yeast[43,44]. Micro-C has recently been adapted for use in mammalian genomes and has demonstrated a better signal-to-noise ratio compared to Hi-C and the potential to generate contact matrices at very high resolution[45–47]. However, since Micro-C is a genome-wide approach, its sensitivity and resolution are limited by sequencing depth. We have recently developed a 3C-based method that combines MNase digestion with an oligonucleotide-based enrichment approach and enables deep, targeted sequencing of chromatin interactions with viewpoints of interest[48]. This Micro-Capture-C (MCC) approach can generate 3C interaction profiles at base-pair resolution. Here, we present an extension of the MCC method, based on the integration of an enrichment strategy that involves dense tiling of capture oligonucleotides across large regions of interest. This Tiled-MCC approach allows for the generation of regional contact matrices at very high resolution (up to 20 bp) and can therefore resolve local nano-scale chromatin structures with unprecedented detail.

We have used Tiled-MCC to study the effects of cohesin and CTCF depletion in four well-characterized gene loci in mouse embryonic stem (mES) cells. We show that cohesin depletion in these regions leads to a subtle reduction in enhancer–promoter interactions, which is associated with a small decrease in gene expression. CTCF depletion does not generally reduce interactions between the investigated enhancers and promoters, but can lead to rewiring of interactions and ectopic gene activation in some cases. At the level of nano-scale structures, we find that CTCF mediates local insulation within *cis*-regulatory elements, but that cohesin does not strongly contribute to these localized interaction patterns. Together, our findings provide insights into the role of loop extrusion in regulating fine-scale chromatin architecture and enhancer–promoter interactions.

## Results

**Tiled-MCC can generate local contact matrices at high resolution.** We have recently developed the MCC approach, which is capable of generating base-pair resolution 3C data for selected viewpoints[48]. To achieve this, MCC combines the following features: (1) permeabilization of cells using digitonin instead of traditional detergents, thus maintaining cellular and nuclear architecture; (2) digestion with MNase instead of restriction enzymes; (3) high library complexity and efficient enrichment of viewpoints of interest, thus allowing for deep, targeted sequencing; (4) direct sequencing and bioinformatic reconstruction of ligation junctions.

However, since the MCC viewpoints are narrow (120 bp), MCC data do not define complete 3D structures of regions of interest. We have therefore extended the MCC approach and combined it with an enrichment strategy based on capture oligonucleotides tiled across large genomic regions of interest (Fig. 1a). For the development of this Tiled-MCC approach, we designed panels of biotinylated oligonucleotides optimized for hybridization to MNase-digested 3C libraries. Since MNase cuts the genome largely independent of sequence, the panels are designed to densely cover regions of interest with oligonucleotides of 70 nucleotides in length and an overlap of 35 nucleotides. This results in very efficient enrichment with about ~80% of reads on target. Together, these improvements allow for the generation of local contact matrices at high resolution with high reproducibility (Fig. 1, Supplementary Fig. 1, Supplementary Note 1).

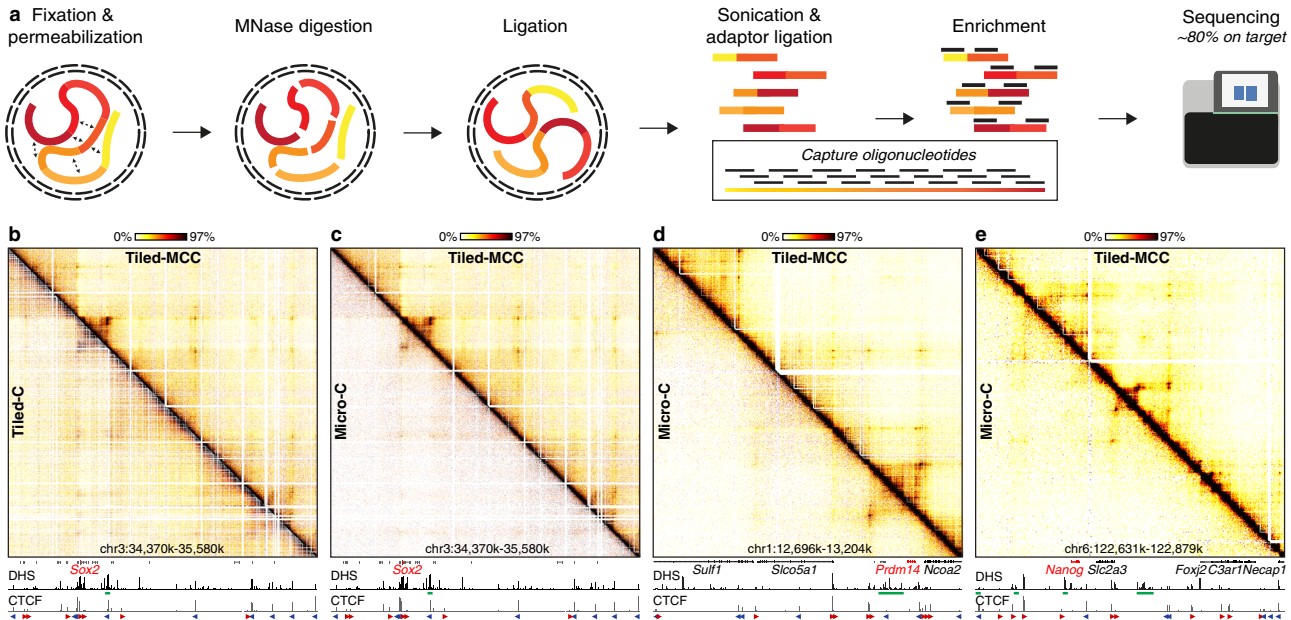

**Fig. 1 Tiled-MCC generates local contact matrices with high sensitivity and resolution. a** Overview of the Tiled-MCC procedure. Cells are initially crosslinked with formaldehyde and permeabilized with digitonin. The chromatin is subsequently digested with MNase, which is followed by proximity ligation. After DNA extraction, the MCC libraries are sonicated and ligated with indexed sequencing adaptors. Multiplexed libraries are subsequently enriched for regions of interest using panels of biotinylated capture oligonucleotides, which are optimized for hybridization to MNase-digested 3C libraries, and sequenced. **b** Comparison of Tiled-MCC (top-right) and Tiled-C (bottom-left) contact matrices at 500 bp resolution of the *Sox2* locus in mES cells. **c**–**e** Comparison of Tiled-MCC (top-right) and Micro-C (bottom-left) contact matrices at 500 bp resolution of the *Sox2* (**c**), *Prdm14* (**d**), and *Nanog* (**e**) loci in mES cells. Gene annotation (genes of interest in red, coding genes in black, non-coding genes in gray), DNase hypersensitive sites (DHS), and ChIP-seq data for CTCF are shown below the matrices. The axes of the DHS and CTCF ChIP-seq profiles are scaled to signal and have the following ranges; DHS: *Sox2* = 0–4.46, *Prdm14* = 0–6.45, *Nanog* = 0–10.25; CTCF: *Sox2* = 0–1833, *Prdm14* = 0–2168, *Nanog* = 0–3092. Enhancers of interest are indicated in green below the DHS profiles. The orientations of CTCF motifs at prominent CTCF-binding sites are indicated by arrowheads (forward orientation in red; reverse orientation in blue).

The principle of Tiled-MCC is similar to existing 3C approaches that use large enrichment panels, such as Tiled-C[49], Capture Hi-C[8,50], and Targeted Chromatin Capture[51]. However, an important difference is that Tiled-MCC is based on MNase digestion, whereas the existing approaches use restriction enzymes for chromatin digestion. Approaches based on restriction enzyme digestion are limited in their resolution by the distribution of the recognition motifs in the genome, because the ligation junctions (based on which chromatin interactions are identified) can only be formed at sites where the chromatin has been cut. For enzymes with a 4 bp recognition motif (such as DpnII), the theoretical resolution limit is 256 bp; for enzymes with a 6 bp recognition motif (such as HindIII), the resolution is theoretically limited to 4096 bp. However, due to the irregular distribution of the recognition motifs, bin sizes of 1–2 kb and 5–10 kb, respectively, are usually required to avoid "empty bins" for which information on chromatin interactions is missing due to the absence of restriction sites. Since MNase digests the genome largely independent of sequence, ligation junctions can be formed more uniformly across the genome. MNase digestion can therefore overcome the barrier in resolution imposed by restriction enzymes. Consequently, Tiled-MCC is able to generate contact matrices at a higher resolution compared to Tiled-C (Fig. 1b, Supplementary Fig. 2).

To enable data generation at higher resolution, Tiled-MCC requires deeper sequencing compared to Tiled-C (Supplementary Fig. 3). Since ligation junctions in Tiled-MCC can be formed at any sequence across the genome, a more complex enrichment strategy and deeper sequencing are needed to generate the same number of unique ligation junctions compared to Tiled-C. In contrast, in Tiled-C, capture oligonucleotides can be targeted to

restriction sites. This reduces the number of reads required but limits the resolution to the size of the restriction fragments. In addition, ligation of MNase-digested libraries is less efficient compared to libraries digested with restriction enzymes (Supplementary Note 1). To account for these factors, we have sequenced the Tiled-MCC libraries about 10-fold deeper compared to Tiled-C libraries (Supplementary Table 1). However, down-sampling analyses show that 2–4-fold deeper sequencing compared to Tiled-C suffices for data visualization at 500 bp resolution (Supplementary Figs. 4, 5). This equates to 100–200 million reads per enriched Mb.

Overall, the combination of the advantages of tiled enrichment approaches and the MNase-based MCC approach, makes Tiled-MCC a useful approach for the generation of local contact matrices at very high resolution. Tiled-MCC data can be generated at greater depth and for relatively low sequencing costs compared to genome-wide methods, such as Micro-C (Supplementary Table 1). Using the Tiled-MCC approach, we have generated detailed contact matrices of well-characterized gene loci in mES cells. A comparison of Tiled-MCC and Micro-C[46] data show that Tiled-MCC is able to detect enhancer–promoter interactions and long-range contacts between CTCF-binding elements which are not detected by Micro-C (Fig. 1c–e, Supplementary Table 2).

**Tiled-MCC enables analysis of nano-scale chromatin interactions.** In the Tiled-MCC approach, ligation junctions are sequenced directly, because the libraries are sonicated to an average size of ~200 bp and sequenced with paired-end reads of 150 bp each. The data are analyzed with a pipeline in which the exact positions of the junctions are reconstructed[48]. This allows

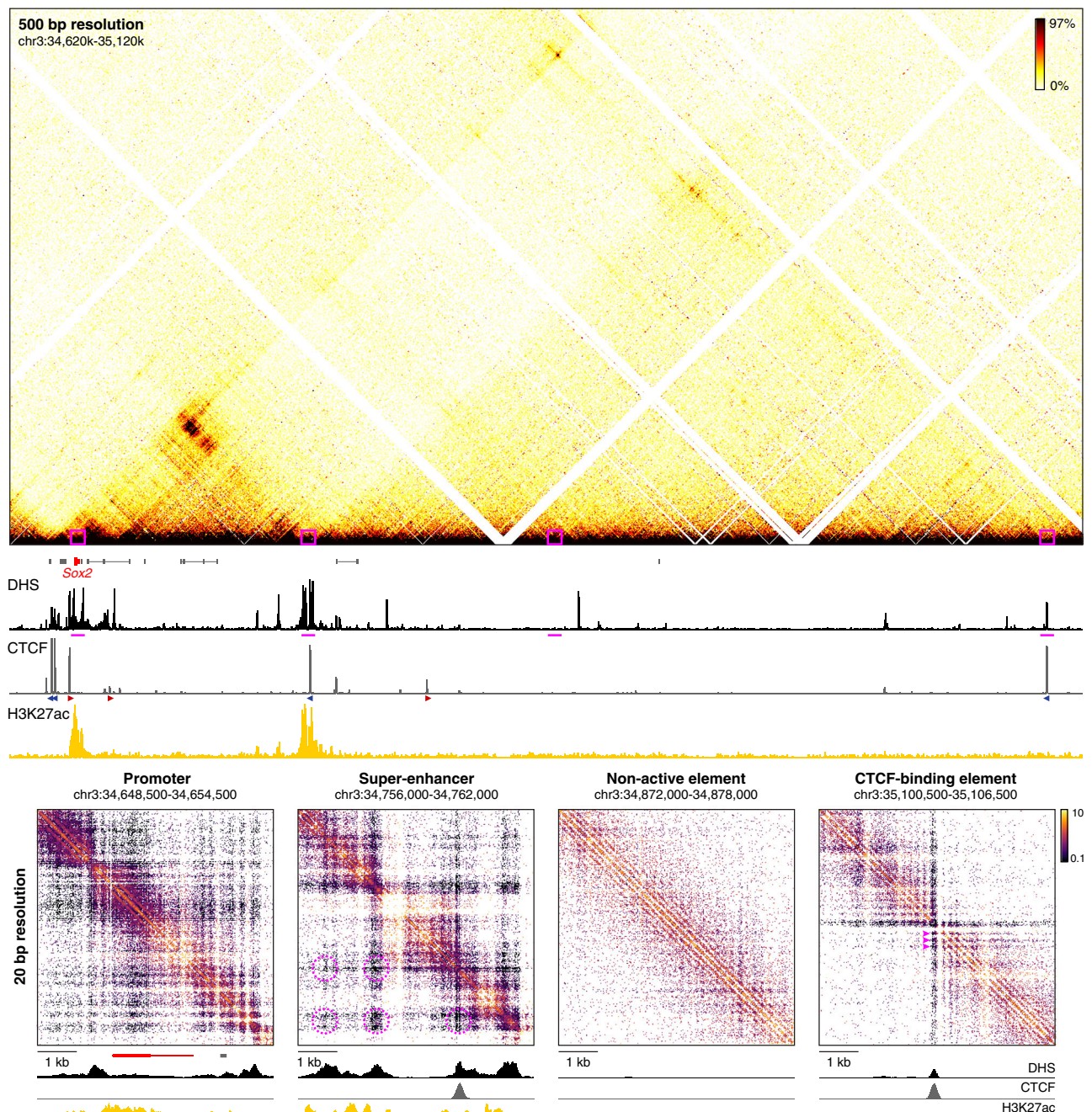

**Fig. 2 High-resolution analysis of Tiled-MCC ligation junctions identifies micro-topologies of *cis*-regulatory elements in the *Sox2* locus.** The fine-scale contact matrices at the bottom show ligation junctions identified by Tiled-MCC in the *Sox2* locus at 20 bp resolution. A large-scale contact matrix (500 bp resolution), gene annotation (*Sox2* in red, coding genes in black, non-coding genes in gray), DNase hypersensitive sites (DHS), and ChIP-seq data for CTCF and H3K27ac for the extended *Sox2* locus are shown in the top panels. The 6 kb regions covered in the fine-scale contact matrices are highlighted with magenta boxes in the contact matrix at the top and with magenta bars below the top DHS profile, and show a promoter, super-enhancer, non-active element, and CTCF-binding element. The axes of the top and bottom DHS and ChIP-seq profiles for CTCF and H3K27ac are fixed and have the following ranges: DHS = 0–5; CTCF = 0–1500; H3K27ac = 0–50. The orientations of CTCF motifs at prominent CTCF-binding sites are indicated by arrowheads (forward orientation in red; reverse orientation in blue). The magenta highlights in the contact matrix covering the super-enhancer indicate enriched interactions between DHSs; the magenta highlights in the contact matrix covering the CTCF-binding element indicate phased nucleosomes.

for the identification of the precise locations of chromatin interactions, which increases the signal-to-noise ratio of the Tiled-MCC data and allows for local analysis at extremely high resolution. The identified junctions can be visualized in density plots, in which their exact locations are plotted and the density of these plotted points is indicated with a color code (Supplementary Fig. 6). Although these density plots allow for direct visualization

of the precise locations of the junctions, they are difficult to interpret quantitatively and not straightforward to normalize. Traditional ICE-normalized[52] contact matrices at very high resolution (20 bp) uncover the same features without artifacts (Supplementary Fig. 6). These matrices allow for the analysis of local chromatin structures with unprecedented resolution and reveal characteristic micro-topologies across the genome (Fig. 2).

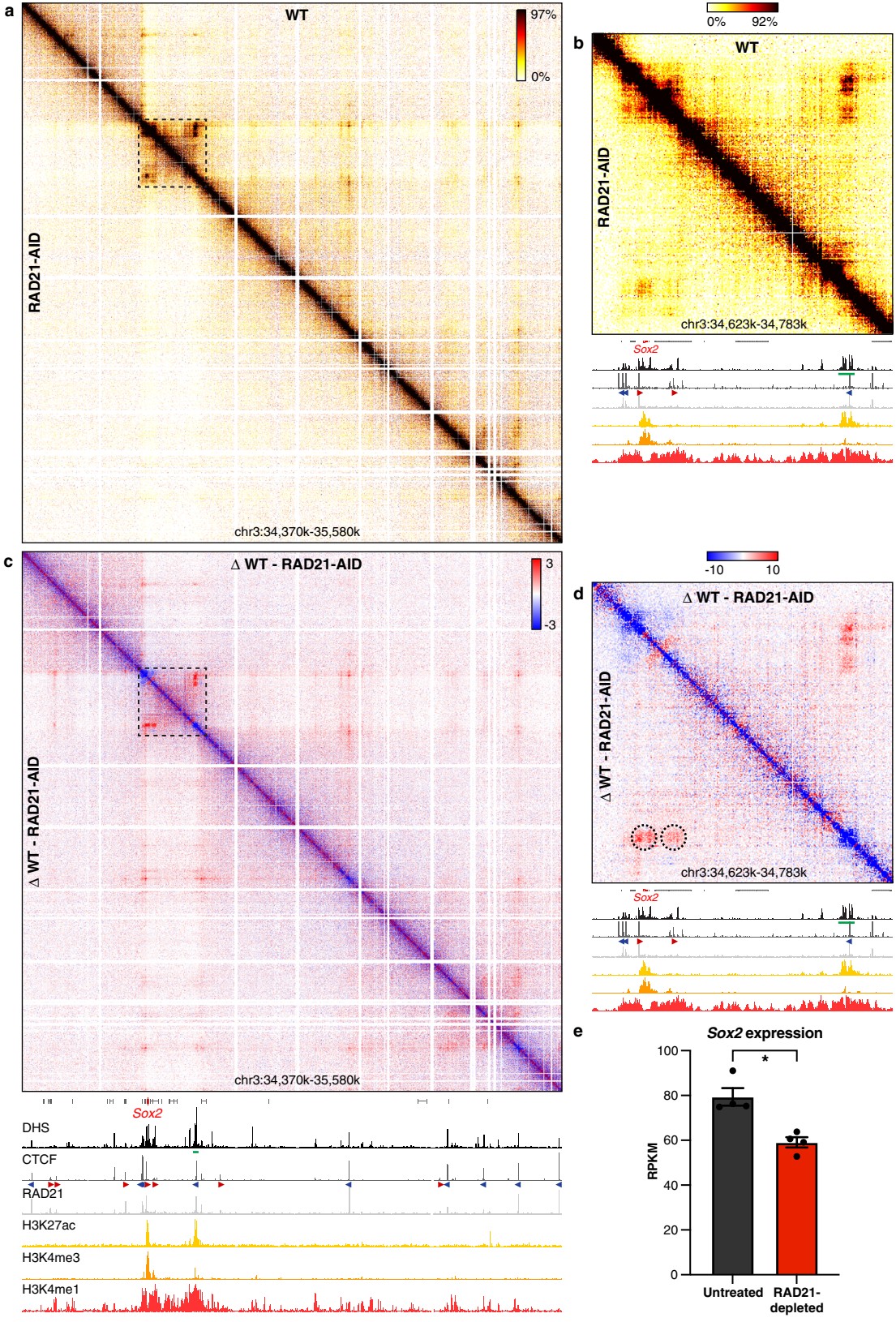

The analysis of micro-topologies requires sequencing of Tiled-MCC data to a depth of at least 100–200 million reads per enriched Mb (Supplementary Fig. 7). Since methods based on restriction enzyme digestion can only generate ligation junctions at restriction sites, the direct analysis of such junctions is not meaningful, as they merely reflect the distribution of restriction fragments across the genome (Supplementary Fig. 8). Although Micro-C is also based on MNase digestion, the procedure of library preparation and sequencing do not allow for direct identification of ligation junctions (Supplementary Note 1). Because Micro-C ligation junctions are inferred, the resolution of Micro-C is limited to ~200 bp, which makes current Micro-C

**Fig. 3 Cohesin depletion results in reduced enhancer–promoter interactions in the *Sox2* locus. a** Tiled-MCC contact matrices of the *Sox2* locus in wild-type (WT) mES cells (top-right) and auxin-treated RAD21-AID mES cells (bottom-left) at 500 bp resolution. **b** Zoomed view of the dashed region in **a** to highlight the interactions between the *Sox2* promoter and its super-enhancer. **c** Differential contact matrix of the *Sox2* locus in which interactions enriched in WT mES cells are indicated in red and interactions enriched in auxin-treated RAD21-AID mES cells are indicated in blue. **d** Zoomed view of the dashed region in **c** to highlight the interactions between the *Sox2* promoter and its super-enhancer. **a–d** Gene annotation (*Sox2* in red, coding genes in black, non-coding genes in gray), DNase hypersensitive sites (DHS), and ChIP-seq data for CTCF, Cohesin (RAD21), H3K27ac, H3K4me3, and H3K4me1 are shown below the matrices. The axes of the DHS and ChIP-seq profiles below **a** and **c** are scaled to signal and have the following ranges: DHS = 0–4.46; CTCF = 0–1833; RAD21 = 0–3318; H3K27ac = 0–48; H3K4me3 = 0–82; H3K4me1 = 0–1826. The axes of the DHS and ChIP-seq profiles below **b**, **d** have the same ranges as in **a** and **c**, except for the CTCF ChIP-seq profile, which is scaled 0–300. Enhancers of interest are indicated in green below the DHS profiles. The orientations of CTCF motifs at prominent CTCF-binding sites are indicated by arrowheads (forward orientation in red; reverse orientation in blue). Interactions of interest are highlighted with dashed circles in **d**, with the left circle indicating the interactions between the *Sox2* promoter and its super-enhancer, and the right circle indicating the interactions between CTCF-binding sites downstream of the promoter and the super-enhancer, which also contains a CTCF-binding site. These interactions are decreased upon cohesin depletion. **e** Expression of *Sox2* in untreated (left) and auxin-treated (right) RAD21-AID mES cells, derived from RNA-seq data, normalized for reads per kilobase of the transcript, per million mapped reads (RPKM)[29]. The bars represent the average of $n = 4$ replicates and the error bars indicate the standard error of the mean. *$P = 4.46E-07$ (calculated using DESeq2 analysis and adjusted for multiple comparisons, as previously decribed[29]). Source data are provided as a Source Data file.

protocols unsuitable for the detection of very high-resolution features of chromatin structure (Supplementary Fig. 9). Therefore, the analysis of micro-topologies is a unique feature of Tiled-MCC, which uncovers distinct nano-scale interaction patterns at *cis*-regulatory elements which cannot be appreciated with existing approaches.

Micro-topology analysis in the *Sox2* locus shows that the regions containing the promoter and super-enhancer are characterized by very fine-scale compartmentalization (Fig. 2). Within the super-enhancer, we also observe a complex pattern of specific interactions between the individual enhancer elements. The structure of a typical CTCF-binding element in this region is characterized by a distinct arrangement of phased nucleosomes and localized stripes, resulting in strong insulation of the regions up- and downstream of the CTCF-binding site. The CTCF-binding site within the super-enhancer also creates localized insulation between the different fine-scale compartments. Furthermore, non-active chromatin is characterized by regular nucleosome arrangements without specific phasing patterns.

**The effects of cohesin depletion on enhancer–promoter interactions**. The advantages of Tiled-MCC make it a unique approach to investigate changes in local chromatin structure in detail upon depletion of architectural proteins. To study the role of cohesin and CTCF in mediating large- and fine-scale chromatin structure, we used mES cell lines in which the cohesin subunit RAD21[29] (SCC1) or CTCF[27] can be rapidly degraded via an auxin-inducible degron (AID). To examine enhancer–promoter interactions in the absence of cohesin, we produced Tiled-MCC matrices of four well-characterized gene loci in RAD21-AID mES cells. We initially focused our analyses on the region containing the *Sox2* gene and its super-enhancer[53]. Consistent with previous reports[26,28], cohesin depletion results in a near-complete loss of TAD structure and long-range interactions between CTCF-binding sites (Fig. 3a, Supplementary Fig. 10). A close-up view reveals that interactions between the *Sox2* promoter and its super-enhancer remain partially intact, despite loss of the interactions between the CTCF-binding sites surrounding these elements (Fig. 3b). However, the interactions between the enhancers and promoters are reduced in intensity (Fig. 3c, d). This reduction in enhancer–promoter interactions corresponds to a decrease in *Sox2* expression of ~25% (Fig. 3e). In contrast, deletion of the *Sox2* super-enhancer decreases expression of *Sox2* in mES cells by ~90%[53,54]. This suggests that the enhancers are still able to upregulate *Sox2* expression in absence of cohesin, but with lower efficiency.

We next investigated the *Prdm14*, *Nanog*, and *Pou5f1* loci, which are also regulated by well-characterized enhancers. In each of these loci, we find a decrease in enhancer–promoter interactions and gene expression following cohesin depletion (Supplementary Figs. 11, 12). In line with our findings for the *Sox2* locus, the impact of cohesin depletion on gene expression in these loci is smaller than the effects of genetic deletions of the enhancers[55–58] (Supplementary Figs. 11, 12). Together, these results indicate that cohesin contributes to the interactions between enhancers and promoters, but is not solely responsible for mediating these interactions.

**The effects of CTCF depletion on enhancer–promoter interactions**. The effects of cohesin depletion could be a direct result of the loss of the process of active extrusion by the cohesin complex, or be due to the loss of insulation within the TADs that result from loop extrusion. To get further insight into the underlying mechanism, we studied the effects of CTCF depletion, which leads to a loss of extrusion boundaries, but does not interfere with loop extrusion otherwise. Consistent with the literature[27,28], Tiled-MCC data in CTCF-AID mES cells show that CTCF depletion results in a loss of TADs and interactions between CTCF-binding sites (Fig. 4a, b, Supplementary Fig. 13). Furthermore, at the *Nanog* locus, we find that loss of CTCF leads to rewiring of the interactions between enhancers and promoters (Fig. 4a, b). Upon CTCF depletion, the super-enhancer forms stronger contacts with the proximal enhancers and the promoter of the downstream *Foxj2* gene. These increased interactions are associated with upregulation of the expression of *Foxj2* (Fig. 4c). We also observe the formation of ectopic enhancer–promoter interactions at the *Prdm14* locus (Supplementary Fig. 14), which are consistent with previously reported effects of CTCF-binding site deletions in this locus[11,59]. However, CTCF depletion at the *Sox2* and *Pou5f1* loci does not result in significant changes in enhancer–promoter interactions or gene expression in mES cells (Supplementary Figs. 15 and 16). These results show that CTCF is important for the specificity of enhancer–promoter interactions, but not generally required for their formation or maintenance.

**The effects of cohesin and CTCF depletion on nano-scale chromatin structures**. To examine the impact of cohesin and CTCF depletion in further detail, we analyzed the effects on local micro-topologies of *cis*-regulatory elements in selected regions of interest (Fig. 5, Supplementary Fig. 17). As expected, the patterns of phased nucleosomes, stripes, and insulation at CTCF-binding elements are lost upon CTCF depletion. However, surprisingly,

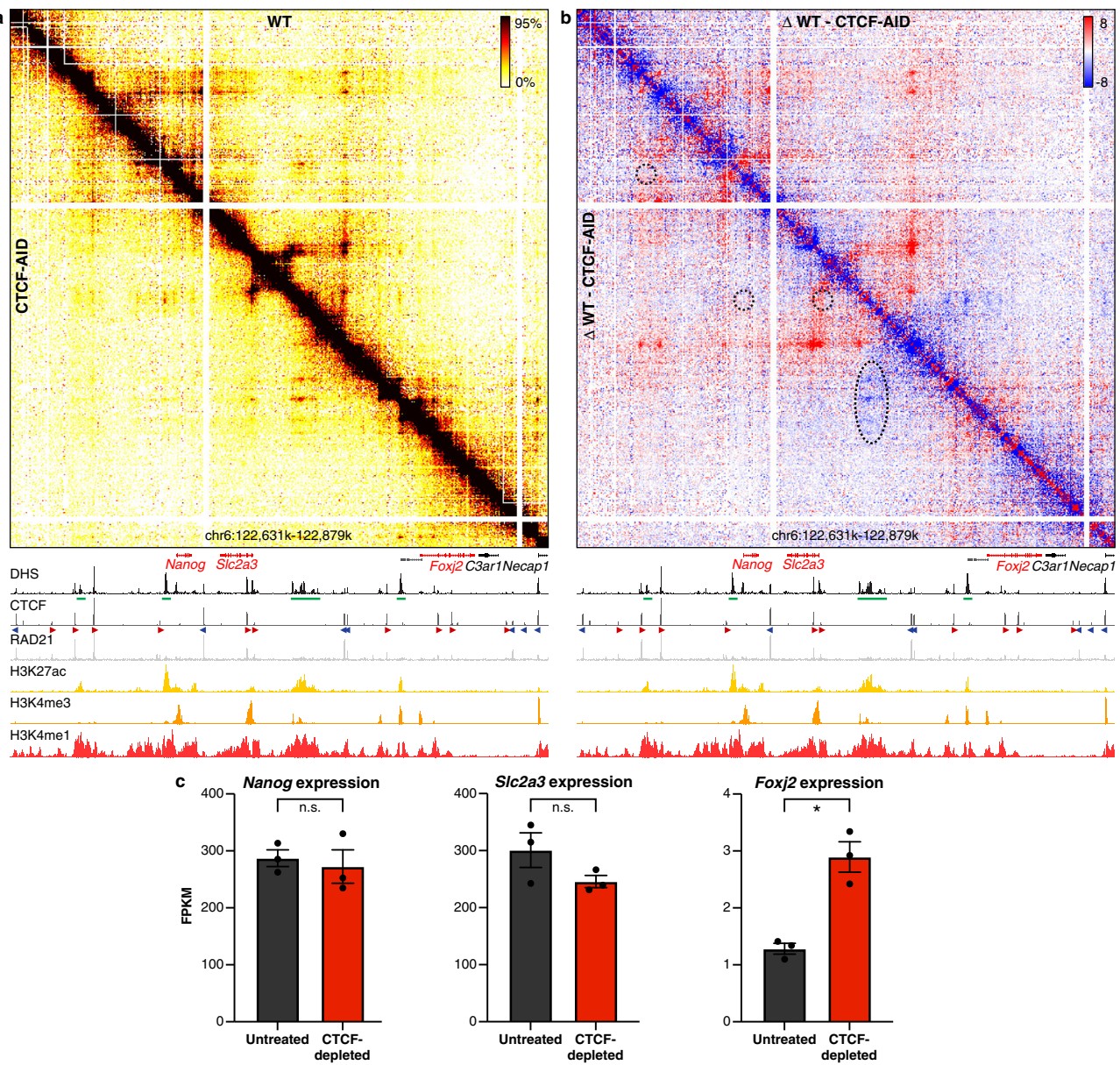

**Fig. 4 CTCF depletion results in ectopic enhancer–promoter interactions in the *Nanog* locus. a** Tiled-MCC contact matrices of the *Nanog* locus in wild-type mES cells (top-right) and auxin-treated CTCF-AID mES cells (bottom-left) at 500 bp resolution. **b** Differential contact matrix of the *Nanog* locus in which interactions enriched in WT mES cells are indicated in red and interactions enriched in auxin-treated CTCF-AID mES cells are indicated in blue. **a**–**b** Gene annotation (genes of interest in red, coding genes in black, non-coding genes in gray), DNase hypersensitive sites (DHS), and ChIP-seq data for CTCF, Cohesin (RAD21), H3K27ac, H3K4me3, and H3K4me1 are shown below the matrices. The axes of the DHS and ChIP-seq profiles are scaled to signal and have the following ranges: DHS = 0–10.25; CTCF = 0–3092; RAD21 = 0–3414; H3K27ac = 0–58; H3K4me3 = 0–90; H3K4me1 = 0–2064. Enhancers of interest are indicated in green below the DHS profiles. The orientations of CTCF motifs at prominent CTCF-binding sites are indicated by arrowheads (forward orientation in red; reverse orientation in blue). The dashed circles and oval in **b** highlight the following interactions of interest (from left to right): *Nanog* promoter and far upstream enhancer; *Nanog* promoter and downstream super-enhancer; *Slc2a3* promoter and downstream super-enhancer; *Foxj2* proximal *cis*-regulatory elements and upstream super-enhancer. The interactions between the *Nanog* and *Slc2a3* promoters with the enhancers in the region appear unchanged upon CTCF depletion (despite the loss of CTCF-mediated interactions directly upstream of the *Slc2a3* promoter), whereas the interactions between the proximal *cis*-regulatory elements of *Foxj2* and the super-enhancer are increased. **c** Expression of *Nanog*, *Slc2a3*, and *Foxj2* in untreated (left) and auxin-treated (right) CTCF-AID mES cells, derived from RNA-seq data, normalized for fragments per kilobase of the transcript, per million mapped reads (FPKM). The bars represent the average of *n* = 3 replicates and the error bars indicate the standard error of the mean. Significant (*) and non-significant (n.s.) changes in expression are indicated. *Nanog*: *P* = 0.7030; *Slc2a3*: *P* = 0.1774; *Foxj2*: *P* = 0.0001 (calculated using Cuffdiff analysis and adjusted for multiple comparisons, as previously decribed[27]). Source data are provided as a Source Data file.

the localized insulating properties are not altered by depletion of cohesin, despite the complete loss of long-range contacts between CTCF-binding sites upon cohesin depletion (Fig. 5, Supplementary Figs. 17, 18). The nano-scale structures of the investigated promoters and super-enhancers are also not grossly affected by

cohesin depletion, but insulation mediated by CTCF-binding sites within the super-enhancers or promoter regions is reduced upon CTCF depletion (Fig. 5, Supplementary Fig. 17). This suggests that CTCF mediates local insulation independent of cohesin-mediated loop extrusion.

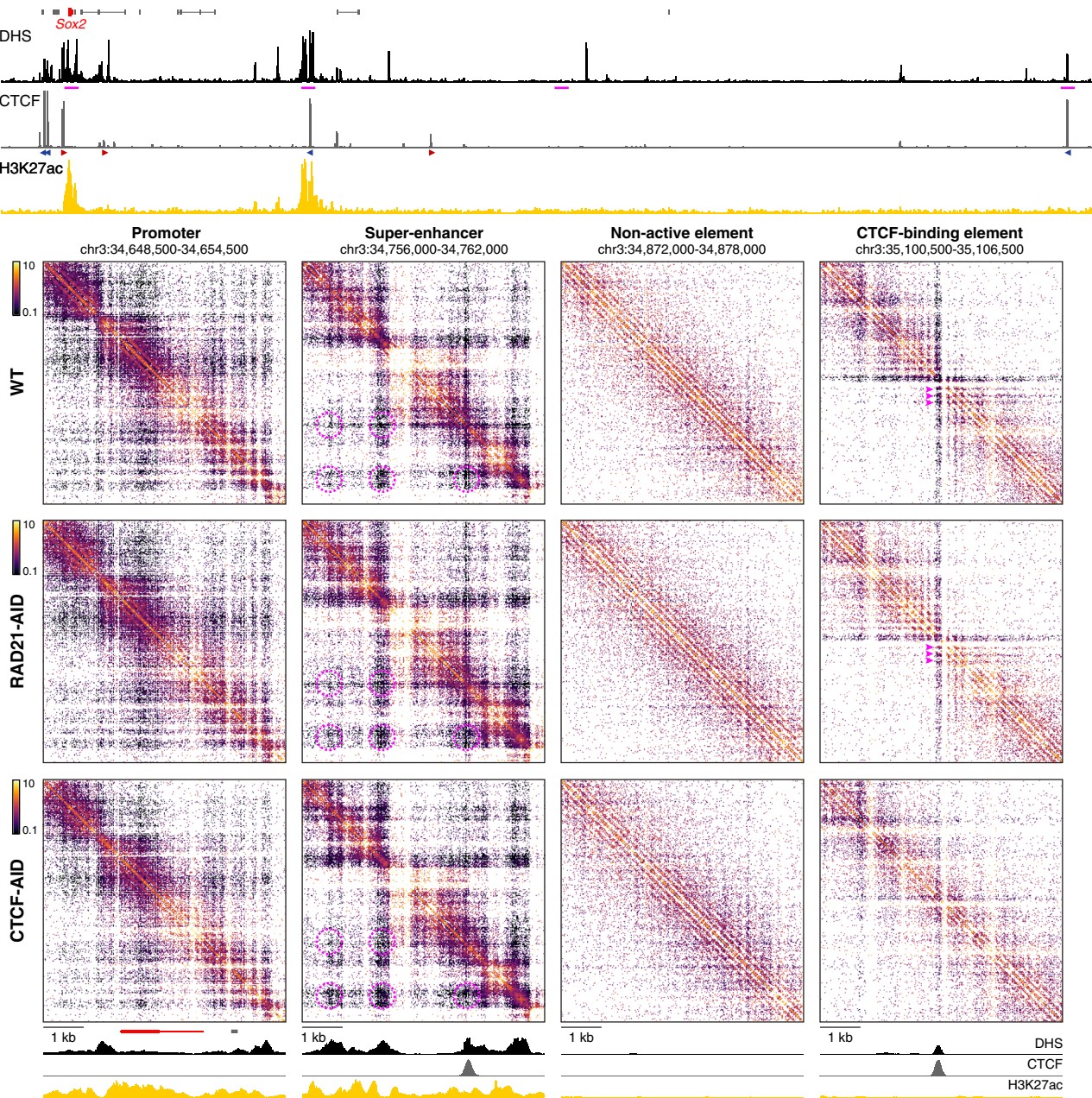

**Fig. 5 Micro-topologies of *cis*-regulatory elements in the *Sox2* locus and their variable dependence on cohesin and CTCF.** The contact matrices show ligation junctions identified by Tiled-MCC in the *Sox2* locus at 20 bp resolution in wild-type (WT), auxin-treated RAD21-AID, and auxin-treated CTCF-AID mES cells. Gene annotation (*Sox2* in red, coding genes in black, non-coding genes in gray), DNase hypersensitive sites (DHS), and ChIP-seq data for CTCF and H3K27ac for the extended *Sox2* locus are shown above the matrices. The 6 kb regions covered in the contact matrices are highlighted with magenta bars below the top DHS profile, and show a gene promoter, super-enhancer, non-active element, and CTCF-binding element. The axes of the top and bottom DHS and ChIP-seq profiles for CTCF and H3K27ac are fixed and have the following ranges: DHS = 0–5; CTCF = 0–1500; H3K27ac = 0–50. The orientations of CTCF motifs at prominent CTCF-binding sites are indicated in the top panel by arrowheads (forward orientation in red; reverse orientation in blue). The magenta highlights in the contact matrices covering the super-enhancer indicate enriched interactions between DHSs; the magenta highlights in the contact matrices covering the CTCF-binding element indicate phased nucleosomes.

These observations are confirmed by micro-topology analysis of a larger region at the *Nanog* locus, which contains a promoter, an enhancer, and both a strong and a weak CTCF-binding site (Fig. 6). Upon cohesin depletion, the specific interactions between the CTCF-binding sites are lost, but the stripe pattern at the strong CTCF-binding site remains, whereas CTCF depletion results in the loss of both features. In contrast, the local structures at the enhancer and promoter are not dependent on cohesin and CTCF.

## Discussion

The Tiled-MCC approach has enabled us to define regulatory interactions within regions of interest in unprecedented detail. We have used this approach to study the effects of cohesin and CTCF depletion on fine-scale chromatin architecture and enhancer–promoter interactions in selected genomic regions (Fig. 7).

We have investigated four regions, containing a total of six genes of interest expressed in mES cells (*Sox2*, *Nanog*, *Slc2a3*,

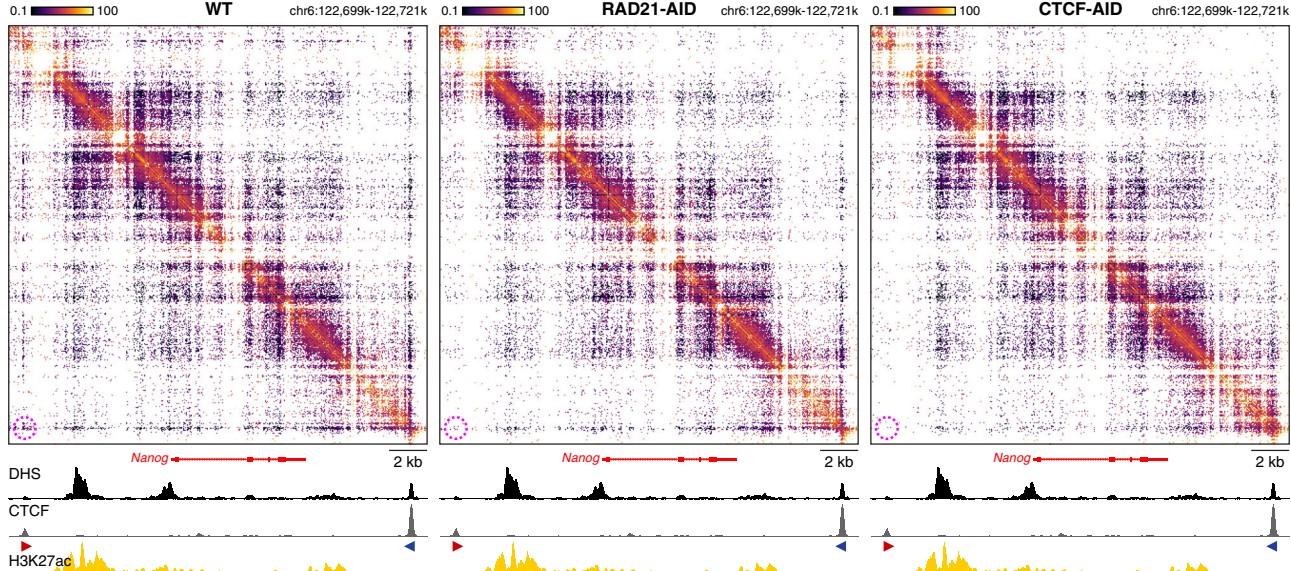

**Fig. 6 Micro-topologies of the *Nanog* locus upon cohesin and CTCF depletion.** The contact matrices show ligation junctions identified by Tiled-MCC in a 22 kb region containing the *Nanog* locus at 50 bp resolution in wild-type (WT), auxin-treated RAD21-AID, and auxin-treated CTCF-AID mES cells. Gene annotation, DNase hypersensitive sites (DHS), and ChIP-seq data for CTCF and H3K27ac are shown below the matrices. The axes of the DHS and ChIP-seq profiles for CTCF and H3K27ac are fixed and have the following ranges: DHS = 0–5; CTCF = 0–1500; H3K27ac = 0–50. The orientations of CTCF motifs at prominent CTCF-binding sites are indicated by arrowheads (forward orientation in red; reverse orientation in blue). The magenta highlights in the contact matrices indicate enriched interactions between CTCF-binding elements in WT mES cells, which are decreased in auxin-treated RAD21-AID and auxin-treated CTCF-AID mES cells.

*Prdm14*, *Slco5a1*, *Pou5f1*). These genes interact with 2–8 individual enhancer elements, resulting in a total of ~30 enhancer–promoter pairs. We find that the strength of the investigated enhancer–promoter interactions is decreased upon cohesin depletion, with the exception of the interactions between the weak enhancer–promoter pairs that span a CTCF boundary (*Nanog* and the downstream super-enhancer and *Slco5a1* and the downstream super-enhancer; Supplementary Figs. 11 and 12). Overall, our data, therefore, show that cohesin depletion results in a reduction of the strength of interactions between enhancers and promoters within TADs in the regions which we investigated.

In contrast, the existing literature suggests that enhancer–promoter interactions are largely maintained upon cohesin depletion. In a previous study, HindIII-based Promoter Capture Hi-C was used to study changes in the chromatin interactions of ~22,000 promoters upon cohesin depletion in HeLa cells[32]. In total, ~120,000 significant interactions were detected, with a resolution of ~4–5 kb. The effects of cohesin depletion could be robustly categorized for 40% of these interactions. The authors report ~36,000 lost interactions, ~13,000 maintained interactions, and ~2,500 gained interactions upon cohesin depletion. The authors show that the lost interactions are enriched for cohesin and CTCF occupancy, whereas the maintained and gained interactions are enriched for chromatin marks associated with active promoters and enhancers. Compared to the Promoter Capture Hi-C data generated in this study, Tiled-MCC has significantly higher resolution and sensitivity and therefore allows for a more detailed quantitative interpretation of interaction strength within a larger dynamic range. This could explain the differences in the observed effects of cohesin depletion on enhancer–promoter interactions. In addition to the Promoter Capture Hi-C study, a recent pre-print based on Micro-C analysis in mES cells has investigated the impact of cohesin depletion on enhancer–promoter interactions and suggested that enhancer–promoter interactions are largely maintained in absence of cohesin[60]. The resolution and sensitivity of Micro-C data are significantly higher compared to Promoter Capture Hi-C data, but lower in comparison to Tiled-MCC data. It is possible

that the relatively subtle decreases in enhancer–promoter interactions require very high sensitivity—as provided by Tiled-MCC—to be detected. Another possible explanation for the discrepancy between the Micro-C and Tiled-MCC data, is that the Tiled-MCC data were generated after 6 h of cohesin depletion, whereas the Micro-C data were generated after 3 h of cohesin depletion.

In addition, the Tiled-MCC data show that CTCF depletion can lead to the rewiring of regulatory interactions. This is consistent with the literature and indicates that TADs contribute to the specificity of enhancer–promoter interactions. In addition, our data show that CTCF depletion does not generally affect the strength of enhancer–promoter interactions within TADs. This is in agreement with Micro-C analyses in mES cells in a recent pre-print[60]. These Micro-C data were generated after 3 h of CTCF depletion, whereas the Tiled-MCC data were generated after 48 h of CTCF depletion; both datasets do not show a general reduction in the strength of enhancer–promoter interactions upon CTCF depletion.

Together, our data demonstrate that CTCF and TAD insulation are not required for enhancer–promoter interactions. Active loop extrusion by cohesin is also not essential for enhancer–promoter interactions but does contribute to their robustness. This suggests that interactions between enhancers and promoters are formed by a different, largely independent, regulatory mechanism, which might involve affinity between the transcription factors and co-factors bound at these elements. Loop extrusion could enhance the interactions between enhancers and promoters, by increasing the proximity between these elements during the process of extrusion or via direct bridging of enhancers and promoters by cohesin. The latter might be particularly important for enhancers and promoters located very close to CTCF-binding sites[30].

It has been suggested that cohesin is particularly important for mediating very long-range interactions between enhancers and promoters[32,33]. While our data show little change in enhancer–promoter interactions across very small distances (<5 kb),

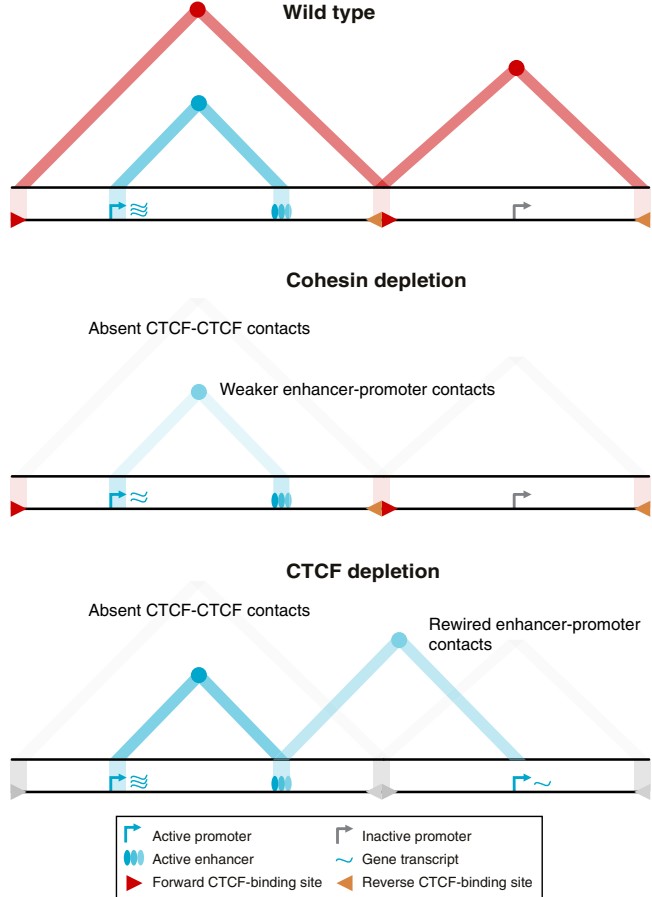

**Fig. 7 Graphical summary.** The top panel shows topologically associating domains (TADs; red triangles), interactions between their CTCF boundaries (red circles at the apexes of the TADs), and enhancer–promoter interactions (blue circle at the intersection between the active promoter and enhancer, as indicated with the light blue triangle) for a hypothetical region of the genome in wild-type cells. Upon cohesin depletion (middle panel), TADs are weakened, interactions between CTCF-binding elements are lost, and interactions between active enhancers and promoters are decreased, causing small changes in gene expression. Depletion of the CTCF protein (bottom panel) causes a loss of TADs and CTCF-mediated interactions but does not generally reduce enhancer–promoter interactions. Depending on the context, however, CTCF depletion can result in the formation of ectopic enhancer–promoter interactions and changes in gene expression.

we observe a decrease in enhancer–promoter interactions and gene expression at the relatively small *Pou5f1* and *Nanog* loci upon cohesin depletion, across distances of ~20 and ~50 kb, respectively.

In our study, we have focused on gene loci that are specifically expressed in mES cells. We find consistent reductions in enhancer–promoter interactions and gene expression upon cohesin depletion in the investigated loci. This appears to be in contrast with reported mild effects on gene expression upon loss of cohesin or NIPBL[25,26]. However, our data suggest that cohesin is particularly important for mediating the robustness of enhancer–promoter interactions. It is therefore perhaps not surprising that only a subset of genes is misregulated upon cohesin depletion, since the loss of cohesin might only affect genes whose expression is critically dependent on the robust formation of tissue-specific enhancer–promoter interactions. This is consistent with the reported importance of cohesin for regulating gene expression upon stimuli and during differentiation[36,37].

In addition to a detailed investigation of the effects of cohesin and CTCF depletion on selected enhancer–promoter interactions, the Tiled-MCC approach has enabled us to identify localized interaction patterns at *cis*-regulatory elements with unprecedented resolution. These analyses reveal distinct nano-scale structures, characterized by very fine-scale compartmentalization and enriched interactions between accessible regions of chromatin within promoters and super-enhancers. Furthermore, we find that binding of the CTCF protein results in strong local insulation in a cohesin-independent manner. This suggests an additional role for CTCF in (local) chromatin organization, which is not mediated via loop extrusion. Further investigation is required to identify the underlying mechanisms, which might involve the recruitment of other factors or the strong impact of CTCF binding on the arrangement of nucleosomes[61].

Taken together, our high-resolution data identify the existence of complex nano-scale structures within *cis*-regulatory elements, which are independent of loop extrusion. Furthermore, we show that loop extrusion is not essential for enhancer–promoter interactions, but an important mechanism to regulate the robustness and specificity of regulatory interactions.

## Methods

**Cell culture**. Wild type (E14), RAD21-mAID-eGFP[29], and CTCF-AID-eGFP[27] mouse embryonic stem (mES) cells were cultured on plates pre-coated with 0.1% gelatin (Sigma-Aldrich, G1393) in Glasgow Modified Essential Medium (Gibco, 11710035) supplemented with 10% fetal bovine serum (Gibco, 10270106), 0.01 mM 2-Mercaptoethanol (Gibco, 31350010), 2.4 mM L-glutamine (Gibco, 25030024), 1× non-essential amino acids (Gibco, 11140050), 1× sodium pyruvate (Gibco, 11360070) and 20 ng/ml recombinant mouse leukemia-inhibitory factor (Cell Guidance Systems, GFM200). Cells were grown at 37 °C and 5% $CO_2$. Half of the media was exchanged daily and cells were passaged every 2 days by trypsinization (Gibco, 25300054).

**Depletion of RAD21 and CTCF**. RAD21-mAID-eGFP and CTCF-AID-eGFP mES cells were passaged and plated 2 days before auxin treatment. On the treatment day, cells were first washed once with phosphate-buffered saline (PBS) and supplemented with an equilibrated (37 °C and 5% $CO_2$) medium containing 500 μM indole-3-acetic acid (Sigma-Aldrich, I5148) freshly prepared before use. RAD21-mAID-eGFP cells were treated with auxin for 6 hr. Since complete depletion of CTCF takes longer to achieve[27,62], CTCF-AID-eGFP cells were treated for 48 hr, with exchanging auxin-containing media after the first 24 hr. Alkaline phosphatase testing for the assessment of pluripotency before and after auxin treatment was performed using the Alkaline Phosphatase Detection Kit (EMD Millipore, SCR004).

**Immunoblotting**. RAD21-mAID-eGFP and CTCF-AID-eGFP mES cells were grown in 25 cm² flasks until they reached ~80% confluency. Cells were then treated with auxin following the aforementioned treatment conditions for each cell line. Whole-cell extracts were prepared by first dissociating cells by trypsinization, resuspending them in media, pelleting, and washing once with PBS. The pellet was then resuspended in 5× the pellet volume in radioimmunoprecipitation assay buffer (Thermo Scientific, 89900) with a protease inhibitor cocktail (Leupeptin, Carl Roth, CN33.4; Pepstatin A, Carl Roth, 2936.3; PMSF, Carl Roth, 6367.3; Benzamidine hydrochloride, Acros Organics, 105245000, resuspended in ethanol) and 250 U/μL benzonase (Sigma-Aldrich, E1014) and rotated for 1 hr at 4 °C. After centrifugation at maximum speed, whole-cell extract in the supernatant was measured using the Bio-Rad Protein Assay (Bio-Rad, 5000006). In all, 15 μg of protein extracts were mixed with NuPAGE LDS Sample Buffer (Invitrogen, NP0007), supplemented with 50 mM DTT (Carl Roth, 6908.3), and loaded on a NuPAGE 4–12% Bis-Tris gel (Invitrogen, NP0321PK2). Proteins were transferred to a PVDF membrane using the XCell Blot Module (Invitrogen, EI9051) for 1 hr at 30 V. Membrane blocking was performed in 5% blotting grade milk powder (Carl Roth, T145.2) dissolved in PBS-0.05% Tween-20 (PBS-T) for 1 hr at room temperature. After rinsing the membranes with PBS-T, incubation with primary antibodies was performed as follows: the membrane was cut into two pieces, one piece corresponding to the higher protein ladder size for the incubation of RAD21 and CTCF primary antibodies (RAD21: anti-RAD21 antibody, rabbit polyclonal, Abcam, ab154769, 1:1000; CTCF: anti-CTCF antibody, rabbit polyclonal, Abcam, ab70303, 1:1000), and the second piece with lower protein ladder size for the incubation with the loading control (Histone H3: anti-histone H3 antibody (HRP), Abcam, ab21054, 1:5000). All primary antibodies were diluted in 2% PBS-T and incubated at 4 °C overnight. The next morning, membranes were washed 3× with PBS-T for 5–10 min each and incubated with secondary antibody (RAD21 and CTCF: Goat Anti-Rabbit IgG H&L (HRP), Abcam ab205718) in PBS-T in a 1:5000

dilution for 1 hr at room temperature, washed again 3× and analyzed on the Intas ChemoCam Imager HR.

**Tiled-MCC—fixation**. The preparation of MCC libraries was performed as previously described[48]. In all, 10 million cells per biological replicate were washed once with PBS, dissociated with trypsin, and resuspended in 10 ml of culture media. Fixation was performed by incubating cells with formaldehyde to a final concentration of 2% on a roller mixer for 10 min at room temperature. The reaction was quenched by adding cold glycine to a final concentration of 130 mM. Samples were centrifuged at 300 rcf for 5 min at 4 °C, the pellet was washed once with cold PBS, centrifuged again, and resuspended in 1 ml of cold PBS containing 0.005% digitonin (Sigma-Aldrich, D141).

**Tiled-MCC—library preparation**. Crosslinked and permeabilized cells were centrifuged at 300 rcf for 5 min. The supernatant was carefully removed without disrupting the pellet, which was subsequently resuspended in 900 µl nuclease-free water. Cells were then split equally into three digestion reactions, such that each reaction contained 3–4 million cells. Titration of different MNase concentrations (NEB, M0247) was performed for each aliquot with MNase concentrations ranging from 30–60 Kunitz U in a total reaction volume of 800 µl containing low-calcium MNase buffer (50 mM Tris-HCl pH 7.5, 1 mM CaCl$_2$). The reaction was then incubated for 1 hr at 37 °C on an Eppendorf Thermomixer at 550 rpm, after which it was quenched with 5 mM of ethylene glycol-bis(2-aminoethylether)-N,N,N′,N′-tetraacetic acid (EGTA, Sigma, E3889). The quenched reaction was subsequently centrifuged for 5 min at 300 rcf and the supernatant was carefully discarded. The pellet was resuspended in 1 ml of PBS with 5 mM EGTA. 200 µl was removed as a control for MNase digestion, from which DNA was extracted using the DNeasy Blood and tissue kit (Qiagen, 69504) and digestion efficiency was assessed using the Agilent D1000 TapeStation (Agilent Technologies, 5067–5582). MNase reactions that yielded ~180 bp fragments, corresponding to mono-nucleosomes with linkers, were taken further for the subsequent reactions.

To minimize DNA loss, DNA end-repair, phosphorylation and ligation were performed in a single tube. However, by controlling the temperature at which the respective enzymes are active, end-repair and phosphorylation were performed before ligation. The remaining 800 µl of MNase-digested chromatin resuspended in PBS and EGTA was centrifuged for 5 min at 300 rcf. The supernatant was carefully discarded and the pellet was resuspended in DNA ligase buffer (Thermo Scientific, B69) supplemented with dNTPs (NEB, N0447L) at a final concentration of 400 µM each and 2.5 mM EGTA. T4 Polynucleotide Kinase (NEB, M0201L), DNA Polymerase I Large Fragment (Klenow, NEB, M0210L), and T4 DNA ligase (Thermo Scientific, EL0013) were added to final concentrations of 200 U/ml, 100 U/ml, and 300 U/ml, respectively. The reaction was incubated on an Eppendorf Thermomixer at 550 rpm for 2 hr at 37 °C, followed by a 16-h incubation at 20 °C. Decrosslinking of chromatin was performed using proteinase K (included in the Qiagen DNeasy blood and tissue kit) at 65 °C for >4 hr, which was followed by DNA extraction using the Qiagen DNeasy Blood and tissue kit. The ligation product, referred to as 3C library hereafter, was assessed using the Agilent D1000 TapeStation. A successful ligation is indicated by a significant increase in the fragment size > ~370 bp.

**Tiled-MCC—sonication, and ligation of indexed sequencing adapters**. Sonication of 3C libraries was performed using 3–5 µg per library on a Covaris S220 Focused Ultrasonicator with the following conditions: 250–300 s: duty cycle 10%; intensity 5; cycles per burst 200, to yield an average fragment size of 200 bp. The sonication quality was assessed using the Agilent D1000 TapeStation. The DNA was purified using Ampure XP beads (Beckman Coulter, A63881). To maximize library complexity, the addition of sequencing adapters was parallelized in triplicate reactions such that each reaction contained 1–2 µg of sonicated 3C library. NEB Ultra II (NEB, 7645 S) reagents were used following the manufacturer's protocol with the following deviations: (1) 2–3× the number of adapters was used; (2) all Ampure XP bead clean-up reactions were performed with a DNA sample:bead ratio of 1:1.5; (3) to maximize library complexity and yield, the PCR was performed in triplicate per ligation reaction using the Herculase II PCR reagents (Agilent Technologies, 600677). The parallel library preparations and PCR reactions were subsequently pooled for each reaction.

**Tiled-MCC—capture oligonucleotide design**. Tiled-MCC capture panels were designed to densely cover regions of interest, with oligonucleotides that are 70 nucleotides in length and have an overlap of 35 nucleotides. The sequences were designed and filtered for repetitive sequences using a python-based oligo tool[49] (https://oligo.readthedocs.io/en/latest/). The panels of double-stranded capture oligonucleotides were ordered from Twist Bioscience (Custom probes for NGS target enrichment).

**Tiled-MCC—enrichment**. The enrichment procedure was performed using the Twist Hybridization and Wash Kit (Twist Bioscience, 101025), Twist Universal Blockers (Twist Bioscience, 100578), and Twist Binding and Purification Beads (Twist Bioscience, 100983). Per hybridization reaction, up to 8 amplified and indexed libraries were multiplexed to a final amount of 1.5 µg in a single 0.2-ml

PCR strip-tube. Library pools were dried completely in a vacuum concentrator at 45 °C. Dried DNA was resuspended in 5 µg of mouse Cot-1 DNA and 7 µl of Twist Universal Blockers. In a separate PCR 0.2-ml strip-tube, the probe solution was prepared by mixing 20 µl of Twist Hybridization Mix with 1–2 µl of oligonucleotides and the final volume was adjusted with nuclease-free water. To prepare both the probe solution and the resuspended indexed library pool for hybridization, the probe solution was heated to 95 °C for 2 min in a PCR thermal cycler with the lid at 105 °C, then immediately cooled down on the ice for 5 min. While the probe solution was cooling down on the ice, the library pool was heated following the same conditions for 5 min. After equilibrating both mixtures to room temperature for 1–2 min, the probe solution was added to the library pool, and 30 µl of Twist Hybridization Enhancer was added last. The capture reaction was incubated at 70 °C for 16 hr in a PCR thermal cycler with the lid heated to 85 °C. The hybridization reaction was subsequently mixed with Streptavidin Binding Beads for 30 min at room temperature on a shaker. Washing with Twist Wash Buffers 1, 2, and 3 was performed following the Twist Target Enrichment Protocol. Post-hybridization PCR was performed with 11–12 amplification cycles. The enriched library was purified with pre-equilibrated Twist DNA Purification Beads at a ratio of 1:1.8 DNA to beads. DNA quantification and QC validation were performed using the Qubit dsDNA Broad Range Quantification Assay (Life Technologies, Q32850) and the Agilent Bioanalyzer Broad Range DNA kit (Agilent Technologies, 5067–5582), respectively.

**Tiled-MCC—Sequencing**. Libraries were sequenced using the Illumina NovaSeq and NextSeq 550 platforms with 150 bp paired-end reads. Depending on the quality of the MCC libraries, sequencing 100–200 million reads per enriched Mb per pooled sample is sufficient for data at 500 bp resolution.

**Tiled-MCC—Analysis**. Tiled-MCC analysis was performed using the MCC pipeline[48] (https://github.com/jojdavies/Micro-Capture-C). The main scripts are available for academic use from the Oxford University Innovation Software Store (https://process.innovation.ox.ac.uk/software/p/16529a/micro-capture-c-academic/1).

Briefly, adapter sequences were removed using Trim Galore (Babraham Institute, v.0.3.1) and paired-end reads were reconstructed using FLASH[63] (v.1.2.11). Reads were then mapped to the DNA sequences corresponding to the enriched regions of interest with the non-stringent aligner BLAT[64] (v.35), using a custom-made file containing the sequences of the regions of interest as reference. Based on the mapping by BLAT, the reads in the FASTQ files were split into two or more reads corresponding to the chimeric fragments, and into different files based on the region, they mapped to, using the MCCsplitter.pl script. Uninformative reads that did not contain a sequence that mapped to any of the enriched regions of interest were discarded. The split reads in the FASTQ files were subsequently mapped to the mm10 reference genome using Bowtie2[65] (v.2.3.5). The aligned reads were further processed using the MCCanalyser.pl script. PCR duplicates were removed based on the sonicated ends and ligation junction and allowing for a wobble of ±2 bp. Unique ligation junctions were identified if the fragment ends were less than 5 bp apart in the original read and separated by mapping with BLAT and Bowtie2. The orientation of the reads was considered to enable precise identification of the ligation junction.

To generate contact matrices, the unique ligation junctions were converted into raw matrices, which were balanced using ICE normalization[52]. The large-scale contact matrices (500 bp resolution) are shown on a linear scale, whereas the fine-scale contact matrices (20–50 bp resolution) are shown on a log scale. To generate density plots, the unique ligation junctions were filtered for a minimum distance ≥10 bp and plotted in python as a scatter plot with a color code defined by the local density of the data points.

The data presented in the manuscript represent the averages of nine replicates for WT samples, four replicates for untreated RAD21-AID samples, six replicates for auxin-treated RAD21-AID samples, and six replicates for auxin-treated CTCF-AID samples. For direct comparisons between Tiled-MCC matrices, the data were downsampled to the lowest number of filtered, unique ligation junctions per condition or replicate. The RAD21-AID and CTCF-AID cell lines are derived from the same strain as the WT cells and contact matrices from the untreated RAD21-AID samples are nearly identical to the WT samples (Supplementary Fig. 10b). The contact matrices presented in the manuscript, therefore, show comparisons between auxin-treated RAD21-AID and CTCF-AID samples and WT samples.

**Micro-C analysis**. The Micro-C data were downloaded from GSE130275[46] as files listing valid pairs in the mm10 reference genome, which were converted into ICE-normalized[52] contact matrices using HiC-Pro[66].

**ChIP-seq analysis**. Alignment to the mm10 reference genome and processing of ChIP-seq data was performed using the NGseqBasic pipeline[67]. ChIP-seq data for CTCF, RAD21, and H3K4me1 were downloaded from GSE30203[68], GSE94452[30], and GSE27844[69], respectively. ChIP-seq data for H3K27ac and H3K4me3 and DNase I hypersensitivity data in mES cells were accessed via ENCODE[70].

**RNA-seq analysis**. The normalized read counts, *P* values, and significance scores for genes of interest in RAD21-AID[29] and CTCF-AID[27] mES cells were extracted from the original processed data files shared by the authors of the respective articles.

**Reporting summary**. Further information on research design is available in the Nature Research Reporting Summary linked to this article.

## Data availability

The data that support this study are available from the corresponding authors upon reasonable request. The Tiled-MCC data generated in this study have been deposited in the NCBI Gene Expression Omnibus under accession code GSE181694. The RNA-seq data in RAD21-AID cells used in this study are available in the ArrayExpress Archive under accession code E-MTAB-7818. The RNA-seq data in CTCF-AID cells used in this study are available in the NCBI Gene Expression Omnibus under accession code GSE98671. The ChIP-seq data for CTCF in mES cells used in this study are available in the NCBI Gene Expression Omnibus under accession code GSE30203. The ChIP-seq data for RAD21 in mES cells used in this study are available in the NCBI Gene Expression Omnibus under accession code GSE94452. The ChIP-seq data for H3K4me1 in mES cells used in this study are available in the NCBI Gene Expression Omnibus under accession code GSE27844. ChIP-seq data for H3K27ac and H3K4me3 and DNase I hypersensitivity data in mES cells were accessed via ENCODE[70]. Source data are provided in this paper.

## Code availability

Scripts for Tiled-MCC analysis are available for academic use through the Oxford University Innovation software store (https://process.innovation.ox.ac.uk/software/p/16529a/micro-capture-c-academic/1).

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

## Acknowledgements

A.M.O. is funded by the Max Planck Society. J.O.J.D. is funded by an MRC Clinician Scientist Award (MRC Clinician Scientist Fellowship MR/R008108). We would like to thank Angelika Feldmann, James Rhodes, and Rob Klose for providing the RAD21-AID mES cell line and Elphège Nora and Benoit Bruneau for providing the CTCF-AID mES cell line. We are grateful to Patrick Cramer for advice and infrastructure support, Michael Lidschreiber for assistance with bioinformatics analysis, Taras Velychko for support with experiments, and Kerstin Maier and Petra Rus for help with sequencing. We would like to thank Douglas Higgs, Jim Hughes, Damien Downes, and Elisa Oberbeckmann for their advice and feedback on the manuscript.

## Author contributions

A.A. performed experiments, analyzed data, and wrote the manuscript. P.H. and M.A.K. performed experiments. K.Q. analyzed data. J.O.J.D. conceived the project, analyzed data, and wrote the manuscript. A.M.O. conceived and supervised the project, performed experiments, analyzed data, and wrote the first draft of the manuscript. All authors edited and contributed to the manuscript.

## Funding

## Competing interests

J.O.J.D. is a co-founder of Nucleome Therapeutics and provides consultancy to the company. The remaining authors declare no competing interests.
