## [Peer Review File · Nature Communications]

Editorial Note: This manuscript has been previously reviewed at another journal that is not operating a transparent peer review scheme. This document only contains reviewer comments and rebuttal letters for versions considered at Nature Communications

REVIEWER COMMENTS

Reviewer #1 (Remarks to the Author):

The authors addressed most of my concerns but there are a few items that need to be addressed better.

I previously mentioned and asked (point 3): 'To better understand the benefits of Tiled-MCC over existing technologies the authors should compare Tiled-MCC to the latest capture-C methods (that use 4-cutters, with probes selectively positioned at the restriction sites). Comparison should be done at equal sequence depth/complexity, etc'.

I do appreciate the added comparisons between Tiled-MCC and Tiled-C provided in new Suppl. Figure 1, as well as the comparison of their sequencing depths in new Suppl Table 2. However, it is not clear from new Suppl Figure 1 whether this comparison was done at equal sequence depth (needed for fair comparison).

A. Thus, comparison in Suppl. Figure 1 needs to be done at equal sequence depth. If this was done already, it needs to be specified in the legend of Suppl. Figure 1.

This concern also comes from Suppl Table 2 which shows that the authors sequenced their Tiled-MCC libraries 10x deeper than their Tiled-C libraries. Yet, the number of valid interactions per bp is almost equal (5.26 vs 3.66)! How is this possible? Is Tiled-MCC indeed superior over Tiled-C, as the authors claim?

B. Authors, please be transparent and explain and discuss this better in the main text. Rather than making an extensive comparison in the main text between Tiled-MCC and Micro-C (obviously, Tiled-MCC, a targeted approach, provides superior local resolution....) the authors need to convincingly show and discuss that Tiled-MCC outperforms Tiled-C in resolution and performance.

Supplementary Figure 1 (comparison between Tiled-MCC and Tiled-C) shows that Tiled-MCC contact matrices are more crisp than Tiled-C matrices (provided they come from similarly sized datasets, see point A), particularly at higher resolution (500bp instead of 2000bp). Yet, I don't appreciate the discovery of new structures by Tiled-MCC (at 500bp resolution) that were undetectable by Tiled-C (at 2000bp resolution)?

C. What do the micro-topologies shown in the main figures look like if Tiled-C was used and sequenced to identical depth? Are these structures only to be discovered by Tiled-MCC, or is it mostly sequencing depth that matters and are they also appreciable by Tiled-C? Perhaps appreciable at 2000bp resolution?

In summary, the authors should convincingly show the benefits of Tiled-MCC over Tiled-C.

In my original report I mentioned: 'I have difficulties understanding what these 'nano-structures' may be, and whether they reflect chromatin topology, protein occupancy or perhaps even artifacts induced by the method. For example: why is coverage along the diagonal so unequal? Is this because MNase protected sequences contribute more reads to the contact matrix than MNase sensitive sequences (protein occupancy)?' The authors understand this concern and have now replaced all density plots in all figures with ICE-normalized contact matrices.

D. Authors, please spend a Suppl. Figure on this topic, showing the reason for and consequences of this transformation as you now explain and show to me in your rebuttal. This is important information for researchers who want to apply Tiled-MCC.

I remain puzzled about the biological significance of 'nano-scale structures'.

Reviewer #2 (Remarks to the Author):

I would like to thank the authors for their efforts to address my concerns and suggestions. Unfortunately, my impression remains that the novel insights from the reported findings remain very limited, and as detailed below I don't think are always discussed adequately in the context of the existing literature. Therefore, the strengths of this manuscript are for me in the development of the MCC technology to Tiled-MCC for Mb-sized continuous genomic regions, which will be a very welcome addition to the experimental toolbox for many researchers in the 3D genome community.

Specific points:

1. Lines 78-82: "Identifying the (potential) roles of cohesin and CTCF in the formation of fine-scale chromatin structures – including the specific regulatory interactions between enhancers and promoters which control gene expression – would contribute to a better understanding of the roles of loop extrusion and TADs in gene regulation and the relationship between genome structure and function." My understanding was that some of the approaches used, most notably Promoter Capture Hi-C as in Thiecke et al 2020, do offer the resolution and sensitivity to identify statistically significant interactions between promoters and enhancers. Do the authors agree with this? I understand that the resolution of Hi-C and Promoter Capture Hi-C is limited by the length of restriction fragments generated by the enzyme used, and that the resolution is greatly increased by MCC and Tiled MCC. But to state that other approaches cannot reliably and reproducibly identify interactions between promoters and enhancers is misleading, and in my opinion not correct.

2. Lines 151-153: "The direct identification of ligation junctions increases the signal-to-noise ratio of the Tiled-MCC data and allows for generation of localized contact matrices at very high resolution (20 bp)..."

One of the major advances of MCC was its base-pair resolution – why is the resolution of Tiled MCC 20 bp and not below? Is it because, according to Supplementary Table 2, MCC was sequenced ~4 times deeper? And how does the resolution change with increasing sequencing read depth (see also point below)? These are important considerations for readers who will consider using this technology.

3. Comparison between Tiled-MCC and Tiled-C: according to Supplementary Table 2, the sequencing depth for Tiled-MCC in the Sox2 locus was ~10 times higher than for Tiled-C (1.2 billion compared to 124 million). This results in an increase of the valid interactions per bp from 3.66 in Tiled-C to 5.26 in Tiled-MCC. Is this good value for money? It would be interesting to see a direct comparison with comparable sequencing depth.

4. Lines 246-250: "In contrast, previous studies have concluded that enhancer-promoter interactions are maintained upon cohesin depletion. For example, HindIII-based Promoter Capture Hi-C experiments, which have a resolution of ~5 kb, have suggested that a subset of interactions with gene promoters do not change upon cohesin depletion."

This is contradictory/misleading. The first sentence makes a general statement that previous studies have shown that enhancer-promoter contacts are maintained upon cohesin depletion. Instead of reinforcing this statement, the second sentence then states that this is true for only a subset of promoter-enhancer interactions. So - some change (which includes weakening of course as observed in this study) and some don't.

5. Related to the point above, in the rebuttal the authors state that Thiecke et al. conclude that enhancer-promoter interactions are maintained upon cohesin depletion.

I have just checked the Thiecke et al paper in question again, and it offers a far more nuanced view (the title already implies that much). This statement in its bold form is simply wrong.

In general, I would urge the authors to make a stronger effort to discuss their findings in more depth in the context of the existing literature. I completely agree that Tiled-MCC offers an impressive increase in resolution that will be welcome as a technical development by the 3D genome community. But it comes at the expense of a limited number of interrogated genomic loci in this study (in total ~ 0.1% of the mouse genome). A direct comparison to the Promoter Capture Hi-C study should cover the following: how many promoter-enhancer contacts were interrogated in that study compared to

this manuscript? How many are changed and how many unaffected in both studies by cohesin and CTCF depletion?

6. On the subject of discussing the findings in the context of the existing literature, I think it is worth pointing out that the bioarchive preprint cited (Hsieh et al) specifically studies the short-term maintenance of enhancer-promoter interactions (3 hours). The authors of the Hsieh et al study emphasise that their finding (that CTCF and cohesin are not required for short-term maintenance of E-P interactions) is fully consistent with a role in 'long-term' maintenance and/or establishment for these factors.

Are the time scales comparable (3h) to this study? My understanding is that cohesin depletion was for 6 hours in this study, and CTCF depletion for 48 hours? If this is correct, can the authors comment on the effects of these different time scales on the results, especially in light of the major model proposed in the Hsieh et al study ("time-buffering model").

Minor comments:

Figure 3d Rad12 instead of Rad21

Do the authors think of Figure 7 as a genuine figure or more a graphical summary, as the header suggests?

Minor: Figure 7 the colour of forward and reverse CTCF sites should not change from red/orange to grey in the CTCF depletion as the integrity of these sites is unaffected.

Reviewer #1 (Remarks to the Author):

The authors addressed most of my concerns but there are a few items that need to be addressed better.

We would like to thank the Reviewer for their feedback, based on which we have improved our manuscript. As discussed in detail below, we have added additional analyses to address the Reviewer's concerns relating to the comparison of Tiled-MCC to Tiled-C, sequencing depth, and micro-topology analyses.

I previously mentioned and asked (point 3): 'To better understand the benefits of Tiled-MCC over existing technologies the authors should compare Tiled-MCC to the latest capture-C methods (that use 4-cutters, with probes selectively positioned at the restriction sites). Comparison should be done at equal sequence depth/complexity, etc'.

I do appreciate the added comparisons between Tiled-MCC and Tiled-C provided in new Suppl. Figure 1, as well as the comparison of their sequencing depths in new Suppl Table 2. However, it is not clear from new Suppl Figure 1 whether this comparison was done at equal sequence depth (needed for fair comparison).

A. Thus, comparison in Suppl. Figure 1 needs to be done at equal sequence depth. If this was done already, it needs to be specified in the legend of Suppl. Figure 1. This concern also comes from Suppl Table 2 which shows that the authors sequenced their Tiled-MCC libraries 10x deeper than their Tiled-C libraries. Yet, the number of valid interactions per bp is almost equal (5.26 vs 3.66)! How is this possible? Is Tiled-MCC indeed superior over Tiled-C, as the authors claim?

We agree with the Reviewer that the difference in sequencing depth between Tiled-MCC and Tiled-C would benefit from further explanation. We have added a direct comparison of Tiled-MCC and Tiled-C at equal sequencing depth in Supplementary Figure 3 and added the following statement in the text:

"To enable data generation at higher resolution, Tiled-MCC requires deeper sequencing compared to Tiled-C (Supplementary Figure 3). Since ligation junctions in Tiled-MCC can be formed at any sequence across the genome, a more complex enrichment strategy and deeper sequencing are needed to generate the same number of unique ligation junctions compared to Tiled-C. In contrast, in Tiled-C, capture oligonucleotides can be targeted to restriction sites. This reduces the number of reads required, but limits the resolution to the size of the restriction fragments. In addition, ligation of MNase-digested libraries is less efficient compared to libraries digested with restriction enzymes (Supplementary Note). To account for these factors, we have sequenced the Tiled-MCC libraries about 10-fold deeper compared to Tiled-C libraries (Supplementary Table 1). However, down-sampling analyses show that 2-4-fold deeper sequencing compared to Tiled-C suffices for data visualization at 500 bp resolution (Supplementary Figures 4, 5). This equates to 100-200 million reads per enriched Mb."

In our previous revision, we had already added the following statement about the ligation efficiency in the Supplementary Note:

"A disadvantage of the Tiled-MCC procedure is that it does not involve specific selection of ligation junctions. As a result, not all sequenced reads are informative. Tiled-MCC libraries therefore require relatively deep sequencing considering the small proportion of the genome that is enriched (Supplementary Table 1)."

(...)

The Tiled-C approach⁴ specifically enriches for ligation junctions, since the oligonucleotides are targeted to restriction enzyme cut sites. In addition, Tiled-C libraries are digested with restriction enzymes, which generate sticky overhangs that can be ligated with very high efficiency. Therefore, the vast majority of Tiled-C reads contain useful ligation junctions. Because the complexity of Tiled-C libraries is also very high, this approach supports deep data generation at very low sequencing costs (Supplementary Table 1). However, since the resolution of Tiled-C is limited by the cut site distribution of restriction enzymes, it does not support data generation beyond a resolution of ~2 kb (Supplementary Figure 2)."

B. Authors, please be transparent and explain and discuss this better in the main text. Rather than making an extensive comparison in the main text between Tiled-MCC and Micro-C (obviously, Tiled-MCC, a targeted approach, provides superior local resolution....) the authors need to convincingly show and discuss that Tiled-MCC outperforms Tiled-C in resolution and performance.

As discussed in response to the previous comment, we have performed additional analyses to compare Tiled-MCC and Tiled-C in more detail. We have added a section to describe this comparison in the main text and also added a panel to one of the main figures (Figure 1b).

Supplementary Figure 1 (comparison between Tiled-MCC and Tiled-C) shows that Tiled-MCC contact matrices are more crisp than Tiled-C matrices (provided they come from similarly sized datasets, see point A), particularly at higher resolution (500bp instead of 2000bp). Yet, I don't appreciate the discovery of new structures by Tiled-MCC (at 500bp resolution) that were undetectable by Tiled-C (at 200bp resolution)?

The discovery of new structures refers to the micro-topology analyses. We have clarified this in the following section:

"The analysis of micro-topologies requires sequencing of Tiled-MCC data to a depth of at least 100-200 million reads per enriched Mb (Supplementary Figure 7). Since methods based on restriction enzyme digestion can only generate ligation junctions at restriction sites, the direct analysis of such junctions is not meaningful, as they merely reflect the distribution of restriction fragments across the genome (Supplementary Figure 8). Although Micro-C is also based on MNase digestion, the procedure of library preparation and sequencing do not allow for direct identification of ligation junctions (Supplementary Note). Because Micro-C ligation junctions are inferred, the resolution of Micro-C is limited to ~200 bp, which makes current Micro-C protocols unsuitable for the detection of very high-resolution features of chromatin structure (Supplementary Figure 9). Therefore, the analysis of micro-topologies is a unique feature of Tiled-MCC, which uncovers distinct nano-scale interaction patterns at cis-regulatory elements which cannot be appreciated with existing approaches."

C. What do the micro-topologies shown in the main figures look like if Tiled-C was used and sequenced to identical depth? Are these structures only to be discovered by Tiled-MCC, or is it mostly sequencing depth that matters and are they also appreciable by Tiled-C? Perhaps appreciable at 2000bp resolution? In summary, the authors should convincingly show the benefits of Tiled-MCC over Tiled-C.

As discussed in response to the previous comment, the analysis of ligation junctions of Tiled-C data is not biologically meaningful, as the junctions reflect the distribution of restriction fragments across the genome. Moreover, due to the barrier in resolution imposed by the distribution of restriction sites across the genome, Tiled-C does not allow for data analysis

beyond a resolution of ~ 2 kb. At 2 kb resolution, very fine-scale features of chromatin structure cannot be appreciated. This is explained in the text which we pasted into our response to the previous comment and can be appreciated from Supplementary Figure 8, which we pasted below:

In addition, we have added Supplementary Figure 7, which shows that features of Tiled-MCC micro-topologies can be appreciated when the Tiled-MCC data are down-sampled to a sequencing depth equivalent to Tiled-C (124 M reads):

However, the fine-scale features are more clearly visible at a sequencing depth of 250/500M reads, which is equivalent to 100-200 million reads per enriched Mb, as stated in the revised text of the manuscript.

In my original report I mentioned: 'I have difficulties understanding what these 'nano-structures' may be, and whether they reflect chromatin topology, protein occupancy or

perhaps even artifacts induced by the method. For example: why is coverage along the diagonal so unequal? Is this because MNase protected sequences contribute more reads to the contact matrix than MNase sensitive sequences (protein occupancy)?’ The authors understand this concern and have now replaced all density plots in all figures with ICE-normalized contact matrices.

D. Authors, please spend a Suppl. Figure on this topic, showing the reason for and consequences of this transformation as you now explain and show to me in your rebuttal. This is important information for researchers who want to apply Tiled-MCC. I remain puzzled about the biological significance of ‘nano-scale structures’.

We agree with the Reviewer that it is useful to include this comparison. We have added Supplementary Figure 6 (pasted below) and a section to the main text to clarify this:

“In the Tiled-MCC approach, ligation junctions are sequenced directly, because the libraries are sonicated to an average size of ~200 bp and sequenced with paired-end reads of 150 bp each. The data are analyzed with a pipeline in which the exact positions of the junctions are reconstructed⁴⁸. This allows for identification of the precise locations of chromatin interactions, which increases the signal-to-noise ratio of the Tiled-MCC data and allows for local analysis at extremely high resolution. The identified junctions can be visualized in density plots, in which their exact locations are plotted and the density of these plotted points is indicated with a color code (Supplementary Figure 6). Although these density plots allow for direct visualization of the precise locations of the junctions, they are difficult to interpret quantitatively and not straightforward to normalize. Traditional ICE-normalized⁵² contact matrices at very high resolution (20 bp) uncover the same features without artefacts (Supplementary Figure 6). These matrices allow for the analysis of local chromatin structures with unprecedented resolution and reveal characteristic micro-topologies across the genome (Figure 2).”

Reviewer #2 (Remarks to the Author):

I would like to thank the authors for their efforts to address my concerns and suggestions. Unfortunately, my impression remains that the novel insights from the reported findings remain very limited, and as detailed below I don't think are always discussed adequately in the context of the existing literature. Therefore, the strengths of this manuscript are for me in the development of the MCC technology to Tiled-MCC for Mb-sized continuous genomic regions, which will be a very welcome addition to the experimental toolbox for many researchers in the 3D genome community.

We would like to thank the Reviewer for their feedback, based on which we have improved our manuscript. As discussed in detail below, we have added additional analyses to address the Reviewer's concerns relating to sequencing depth and included a more elaborate discussion of the existing literature.

Specific points:

1. Lines 78-82: "Identifying the (potential) roles of cohesin and CTCF in the formation of fine-scale chromatin structures – including the specific regulatory interactions between enhancers and promoters which control gene expression – would contribute to a better understanding of the roles of loop extrusion and TADs in gene regulation and the relationship between genome structure and function."

My understanding was that some of the approaches used, most notably Promoter Capture Hi-C as in Thiecke et al 2020, do offer the resolution and sensitivity to identify statistically significant interactions between promoters and enhancers. Do the authors agree with this? I understand that the resolution of Hi-C and Promoter Capture Hi-C is limited by the length of restriction fragments generated by the enzyme used, and that the resolution is greatly increased by MCC and Tiled MCC. But to state that other approaches cannot reliably and reproducibly identify interactions between promoters and enhancers is misleading, and in my opinion not correct.

We do not fully agree that the Promoter Capture Hi-C (PCHi-C) approach offers sufficient resolution and sensitivity to reliably detect enhancer-promoter interactions. PCHi-C generally relies on HindIII digestion. Since HindIII has a 6 bp recognition motif, it will theoretically cut every 4096 bp. Due to the uneven distribution of restriction sites across the genome, HindIII fragments have a size distribution spanning ~1-20 kb. Many of these fragments will contain more than one regulatory element (e.g. an enhancer and a CTCF site); it is therefore not possible to distinguish the contribution of these individual elements to the measured interaction frequency with PCHi-C. In addition, the sensitivity of PCHiC is relatively low, which makes it challenging to examine potential subtle changes in the strength of enhancer-promoter interactions. We have previously re-analyzed PCHi-C data and compared these data to other existing 3C approaches. This has shown that PCHi-C data are not able to identify experimentally validated enhancer-promoter interactions. To illustrate this, we have pasted Figure 1 from Hua et al (Nature, 2021) below. This figure shows that PCHi-C data for the alpha-globin promoter bait (Hba-a2) detects interactions with only 1 out of the 5 well-characterized alpha-globin enhancers in erythroid cells (panel b, enhancer locations indicated with green arrows).

We believe that an extensive comparison of PCHi-C to Tiled-MCC data is beyond the scope of our paper. However, we are happy to modify the sentence highlighted by the Reviewer, and have adapted this as follows:

“To address this, it is important to analyze the effects of cohesin and CTCF depletion on chromatin architecture with approaches that enable analysis at very high resolution and sensitivity and which can detect potentially subtle changes in fine-scale chromatin structures.”

2. Lines 151-153: “The direct identification of ligation junctions increases the signal-to-noise ratio of the Tiled-MCC data and allows for generation of localized contact matrices at very high resolution (20 bp)...”

One of the major advances of MCC was its base-pair resolution – why is the resolution of Tiled MCC 20 bp and not below? Is it because, according to Supplementary Table 2, MCC was sequenced ~4 times deeper? And how does the resolution change with increasing sequencing read depth (see also point below)? These are important considerations for readers who will consider using this technology.

We agree with the Reviewer that it is useful for readers to have a better understanding of the sequencing requirements for the generation of high-resolution Tiled-MCC data. We have included down-sampling analyses in the paper to show how the resolution / sensitivity of the data depend on sequencing depth. These analyses are shown in Supplementary Figures 4, 5, and 7.

The Reviewer is correct that MCC achieves higher resolution compared to Tiled-MCC due to a higher number of sequencing reads relative to the size of the enriched regions. We have clarified this in the Supplementary Note by adding the following statement:

“The resolution of Tiled-MCC is therefore predominantly limited by sequencing depth. Our current dataset supports analysis of localized structures at 20 bp resolution. The MCC approach, which enriches for narrow viewpoints instead of large regions as in Tiled-MCC, supports base-pair resolution analysis, because the ratio of sequencing reads per enriched bp is higher³.”

3. Comparison between Tiled-MCC and Tiled-C: according to Supplementary Table 2, the sequencing depth for Tiled-MCC in the Sox2 locus was ~10 times higher than for Tiled-C (1.2 billion compared to 124 million). This results in an increase of the valid interactions per bp from 3.66 in Tiled-C to 5.26 in Tiled-MCC. Is this good value for money? It would be interesting to see a direct comparison with comparable sequencing depth.

We agree with the Reviewer that the difference in sequencing depth between Tiled-MCC and Tiled-C would benefit from further explanation. We have added a direct comparison of Tiled-MCC and Tiled-C at equal sequencing depth in Supplementary Figure 3 and added the following statement in the text:

“To enable data generation at higher resolution, Tiled-MCC requires deeper sequencing compared to Tiled-C (Supplementary Figure 3). Since ligation junctions in Tiled-MCC can be formed at any sequence across the genome, a more complex enrichment strategy and deeper sequencing are needed to generate the same number of unique ligation junctions compared to Tiled-C. In contrast, in Tiled-C, capture oligonucleotides can be targeted to restriction sites. This reduces the number of reads required, but limits the resolution to the size of the restriction fragments. In addition, ligation of MNase-digested libraries is less efficient compared to libraries digested with restriction enzymes (Supplementary Note). To account for these factors, we have sequenced the Tiled-MCC libraries about 10-fold deeper compared to Tiled-C libraries (Supplementary Table 1). However, down-sampling analyses show that 2-4-fold deeper sequencing compared to Tiled-C suffices for data visualization at 500 bp resolution (Supplementary Figures 4, 5). This equates to 100-200 million reads per enriched Mb.”

In our previous revision, we had already added the following statement about the ligation efficiency in the Supplementary Note:

“A disadvantage of the Tiled-MCC procedure is that it does not involve specific selection of ligation junctions. As a result, not all sequenced reads are informative. Tiled-MCC libraries therefore require relatively deep sequencing considering the small proportion of the genome that is enriched (Supplementary Table 1).

(...)

The Tiled-C approach⁴ specifically enriches for ligation junctions, since the oligonucleotides are targeted to restriction enzyme cut sites. In addition, Tiled-C libraries are digested with restriction enzymes, which generate sticky overhangs that can be ligated with very high efficiency. Therefore, the vast majority of Tiled-C reads contain useful ligation junctions. Because the complexity of Tiled-C libraries is also very high, this approach supports deep data generation at very low sequencing costs (Supplementary Table 1). However, since the resolution of Tiled-C is limited by the cut site distribution of restriction enzymes, it does not support data generation beyond a resolution of ~2 kb (Supplementary Figure 2).”

4. Lines 246-250: “In contrast, previous studies have concluded that enhancer-promoter interactions are maintained upon cohesin depletion. For example, HindIII-based Promoter Capture Hi-C experiments, which have a resolution of ~5 kb, have suggested that a subset of interactions with gene promoters do not change upon cohesin depletion.”

This is contradictory/misleading. The first sentence makes a general statement that previous studies have shown that enhancer-promoter contacts are maintained upon cohesin depletion. Instead of reinforcing this statement, the second sentence then states that this is true for only a subset of promoter-enhancer interactions. So - some change (which includes weakening of course as observed in this study) and some don't.

We agree with the Reviewer that this statement is confusing. We have re-written this section, as explained in further detail in response to the next comment.

5. Related to the point above, in the rebuttal the authors state that Thiecke et al. conclude that enhancer-promoter interactions are maintained upon cohesin depletion.

I have just checked the Thiecke et al paper in question again, and it offers a far more nuanced view (the title already implies that much). This statement in its bold form is simply wrong.

In general, I would urge the authors to make a stronger effort to discuss their findings in more depth in the context of the existing literature. I completely agree that Tiled-MCC offers an impressive increase in resolution that will be welcome as a technical development by the 3D genome community. But it comes at the expense of a limited number of interrogated genomic loci in this study (in total ~ 0.1% of the mouse genome). A direct comparison to the Promoter Capture Hi-C study should cover the following: how many promoter-enhancer contacts were interrogated in that study compared to this manuscript? How many are changed and how many unaffected in both studies by cohesin and CTCF depletion?

We agree with the Reviewer that the paper by Thiecke and colleagues is relevant for our work. However, due to the limited resolution and sensitivity of Promoter Capture Hi-C (PCHi-C) data (as described in response to the first comment), it is not straightforward to directly compare the conclusions of the PCHi-C study to our findings.

Thiecke et al detect a total of 118,074 significant interactions in control HeLa cells. In the cells in which SCC1 (RAD21) is depleted, the authors detect 61,702 interactions. The authors have used ~22,000 baited promoter fragments and thus detect ~3-5 interactions per promoter in these cells. This makes it difficult to interpret the data at individual loci in this study.

The identified interactions are defined as “promoter-interacting regions” (PIRs), which correspond to HindIII fragments with an average size of 4096 bp. The identified PIRs are difficult to interpret, as many contain more than one putative regulatory element due to their relatively large size.

Of the 118,074 identified interactions, the authors find that 36,174 are lost, 12,978 are maintained, and 2,484 are gained upon cohesin depletion (the remaining promoter interactions were not assigned to any category at the degree of confidence defined by the authors).

To overcome the challenges of interpreting these changes at individual loci (as described in response to the first comment), the authors investigate which chromatin signals are enriched in these different categories (see Figure 4). The authors state:

“Both baits and PIRs of lost interactions were selectively enriched for the binding of cohesin and CTCF compared with maintained and gained interactions (...), demonstrating that architectural proteins likely mediate these interactions via their direct binding to the interacting regions in cis (Figure 4C). We also assessed the presence of the active chromatin marks H3K4me1 and H3K4me3 in the same way. Strikingly, we found that these marks were selectively enriched at the PIRs of maintained and gained interactions compared with lost interactions (Figure 4C).

(...)

Taken together, these analyses demonstrate that cohesin-dependent interactions are typically longer range and associate with the binding of cohesin and CTCF in cis, while cohesin-independent interactions are shorter range and associate with features of active promoters and enhancers.”

Based on these analyses, we had previously stated that Thiecke et al conclude that enhancer-promoter interactions are maintained upon cohesin depletion. We agree though that this is an over-simplification and have included a more nuanced discussion of this work in our manuscript, as described further below.

Regarding CTCF, Thiecke et al have not classified the gained, maintained and lost interactions based on enrichment for architectural proteins or active chromatin marks, and we have therefore not added this to our discussion.

“Performing the same analysis for CTCF revealed considerably smaller numbers of significantly affected interactions, both overall (17,645 lost, 13,703 maintained, and 1,663 gained) and on average per promoter (Figure S1C). The less pronounced effects of CTCF compared with cohesin depletion on promoter interactions are consistent with the hypothesis that cohesin is the primary factor in facilitating long-range interactions, while CTCF is often, but not always, required for cohesin positioning on the chromatin. It cannot be ruled out, however, that the observed differences could also be due, at least in part, to the residual levels of CTCF previously detected upon auxin treatment in this system. We, therefore, focused on cohesin depletion for the remainder of the study.”

We would like to note that we find it a bit confusing that Thiecke et al discuss “promoter-interacting regions” in the results section describing the PCHi-C data, but make general claims about “enhancer-promoter interactions” in the abstract, highlights and discussion section of the paper. We cannot be sure, but we think that these claims are based on the last part of the results section, where the authors analyze SLAM-seq data to investigate changes in gene expression. They detect 421 upregulated and 266 downregulated genes in cohesin-depleted cells and state:

“The direction of transcriptional change significantly associated with relative changes in the numbers of connected active enhancers ($p = 0.02$, effect size = 0.74; Figures 5B and S6). (...) Next, we focused on 15 active enhancers that engaged both in lost ($n = 15$) and maintained ($n = 19$) promoter interactions with different genes upon cohesin depletion. Transcription of genes that lost interactions with these enhancers was downregulated in cohesin-depleted cells compared with the genes that maintained interactions with the same enhancers (one sided t test, $p = 0.048$, Figure 5D).”

As we stated before, we think it is difficult to interpret individual loci with PCHi-C data due to the limited resolution and sensitivity. In this case, the authors have defined the interacting HindIII fragments (PIRs) as active enhancers when they contain signal for “at least two out of the three marks H3K4me1, H3K4me3 and H3K27ac”. With this definition, these fragments could also be classified as active promoters, and due to their large size, it is possible that they also contain signal for architectural proteins. The authors have not investigated this systematically, but the example shown in Figure 6 of the paper shows CTCF signal at many of the lost interactions. We are therefore not convinced that these can be interpreted as lost enhancer-promoter interactions and find it difficult to interpret the last section of the paper.

However, we agree with the Reviewer that the unbiased PCHi-C data are relevant to include in the discussion of our paper. We therefore added the following sections in our manuscript:

“We have investigated 4 regions, containing a total of 6 genes of interest expressed in mES cells (Sox2, Nanog, Slc2a3, Prdm14, Slco5a1, Pou5f1). These genes interact with 2–8 individual enhancer elements, resulting in a total of ~30 enhancer-promoter pairs. We find that the strength of the investigated enhancer-promoter interactions is decreased upon cohesin depletion, with the exception of the interactions between the weak enhancer-promoter pairs that span a strong CTCF boundary (Nanog and the downstream super-enhancer and Slco5a1 and the downstream super-enhancer; Supplementary Figures 11, 12). Overall, our data therefore show that cohesin depletion results in reduction of the strength of interactions between enhancers and promoters within TADs in the regions which we investigated.”

“In a previous study, HindIII-based Promoter Capture Hi-C was used to study changes in chromatin interactions with ~22,000 promoters upon cohesin depletion in HeLa cells³². In total, ~120,000 significant interactions were detected, with a resolution of ~4–5 kb (the average size of HindIII fragments is 4096 bp). The effects of cohesin depletion could be robustly categorized for 40% of these interactions. The authors report ~36,000 lost interactions, ~13,000 maintained interactions and ~2,500 gained interactions upon cohesin depletion. The authors show that the lost interactions are enriched for cohesin and CTCF occupancy, whereas the maintained and gained interactions are enriched for chromatin marks associated with active promoters and enhancers. Compared to Promoter Capture Hi-C, Tiled-MCC has significantly higher resolution and sensitivity and therefore allows for more detailed quantitative interpretation of interaction strength within a larger dynamic range. This could explain the differences in the observed effects of cohesin depletion on enhancer-promoter interactions in the Promoter Capture Hi-C and Tiled-MCC data.”

6. On the subject of discussing the findings in the context of the existing literature, I think it is worth pointing out that the bioarchive preprint cited (Hsieh et al) specifically studies the short-term maintenance of enhancer-promoter interactions (3 hours). The authors of the Hsieh et al study emphasise that their finding (that CTCF and cohesin are not required for short-term maintenance of E-P interactions) is fully consistent with a role in ‘long-term’ maintenance and/or establishment for these factors.

Are the time scales comparable (3h) to this study? My understanding is that cohesin depletion was for 6 hours in this study, and CTCF depletion for 48 hours? If this is correct, can the authors comment on the effects of these different time scales on the results, especially in light of the major model proposed in the Hsieh et al study (“time-buffering model”).

We agree with the Reviewer that it is relevant to comment on the difference in time scales. We have added the following two sections to our discussion:

“Another possible explanation for the discrepancy between the Micro-C and Tiled-MCC data, is that the Tiled-MCC data were generated after 6 hours of cohesin depletion, whereas the Micro-C data were generated after 3 hours of cohesin depletion.”

“This is in agreement with Micro-C analyses in mES cells in a recent pre-print⁶⁰. These Micro-C data were generated after 3 hours of CTCF depletion, whereas the Tiled-MCC data were generated after 48 hours of CTCF depletion; both datasets do not show a general reduction in the strength of enhancer-promoter interactions upon CTCF depletion.”

We prefer to not discuss the model proposed by Hsieh and colleagues in detail, as this study has not yet been peer reviewed.

Minor comments:

Figure 3d Rad12 instead of Rad21

We thank the Reviewer for pointing this out and have corrected this spelling mistake.

Do the authors think of Figure 7 as a genuine figure or more a graphical summary, as the header suggests?

We think of this figure as a graphical summary.

Minor: Figure 7 the colour of forward and reverse CTCF sites should not change from red/orange to grey in the CTCF depletion as the integrity of these sites is unaffected.

The reviewer is correct that the integrity of the CTCF sites is unaffected upon depletion of the CTCF protein, but we do think it is more intuitive to make these regions grey in the panel describing the effects of CTCF depletion. We have clarified in the legend that the CTCF protein is depleted to indicate that the grey sites refer to the fact that the CTCF sites are no longer occupied.

REVIEWERS' COMMENTS

Reviewer #1 (Remarks to the Author):

the authors have fully addressed my points.

Reviewer #2 (Remarks to the Author):

I thank the authors for addressing my questions. Unfortunately, I have to say that I am now less convinced about the technological advances that Tiled MCC offers than I was when I initially read the manuscript for the first time. The authors admit, and this has to be recommended, that Tiled MCC has several disadvantages over Capture C or Capture Hi-C approaches, which include more a complex enrichment strategy, less efficient ligation, and fewer informative sequencing reads. The advantage is of course that Tiled-MCC does not rely on restriction sites and that the fragment size in the libraries is more uniform – but this has been known since the original Micro-C paper was published in 2015. So the overall method development advances are limited, given that enrichment of 3C and Hi-C libraries with biotinylated oligos has been published before. And I am yet to be convinced that Tiled MCC enables any biologically meaningful discoveries that would not be possible using a Capture C or Capture Hi-C approach, especially those using 4 bp cutter restriction enzymes or combinations of restriction enzymes (see also below).

Specific comments:

1.) I have some further questions about the resolution:

“However, down-sampling analyses show that 2-4-fold deeper sequencing compared to Tiled-C suffices for data visualization at 500 bp resolution (Supplementary Figures 4, 5).”

“However, since the resolution of Tiled-C is limited by the cut site distribution of restriction enzymes, it does not support data generation beyond a resolution of ~2 kb (Supplementary Figure 2).”

I am not convinced by this. ~500 bp (or even below) resolution is equivalent to what Capture-C or Capture Hi-C approaches provide when using a restriction enzyme with 4 pb recognition site (MboI or DpnII for example). The authors repeatedly use the argument that HindIII cuts every 4096 bp on average. True, but by the same logic, a restriction enzyme with a 4 bp recognition site cuts every 256 bp.

An alternative strategy is to use two restriction enzymes, which is the approach Arima Genomics are taking for their Hi-C and Capture Hi-C, further increasing resolution (see also below). I appreciated that the MCC does not rely on restriction sites and therefore offers a more uniform size representation of DNA regions. But my bet is that the average fragment size using two restriction enzymes (Arima’s approach) is very similar to that in Micro-C libraries.

2.) More on resolution:

“Because Micro-C ligation junctions are inferred, the resolution of Micro-C is limited to ~200 bp, which makes current Micro-C protocols unsuitable for the detection of very high-resolution features of chromatin structure (Supplementary Figure 9).”

“In the Tiled-MCC approach, ligation junctions are sequenced directly, because the libraries are sonicated to an average size of ~200 bp and sequenced with paired-end reads of 150 bp each.”

Is this really a true advantage of Tiled-MCC over Micro-C? Surely, Micro-C libraries could be sonicated to an average size of 200 bp and sequenced with PE 150bp, ensuring that ligation junctions are

sequenced directly?

3.) Comparison to Promoter Capture Hi-C, and again resolution:

"We do not fully agree that the Promoter Capture Hi-C (PChi-C) approach offers sufficient resolution and sensitivity to reliably detect enhancer-promoter interactions. PChi-C generally relies on HindIII digestion. Since HindIII has a 6 bp recognition motif, it will theoretically cut every 4096 bp."

It is true that the majority of existing PChi-C data sets have used HindIII for Hi-C library generation. However, as the authors as experts in the field well know, this is not a defining feature of PChi-C. The restriction enzyme can easily be exchanged for a 4 bp cutter to afford higher resolution (albeit at the expense of deeper sequencing required), as demonstrated for example here:
[https://www.cell.com/cell/pdf/S0092-8674\(21\)00382-2.pdf](https://www.cell.com/cell/pdf/S0092-8674(21)00382-2.pdf)

And indeed HiCap, extremely similar to PChi-C has been established in 2015 using a 4 bp cutter:
<https://genomebiology.biomedcentral.com/articles/10.1186/s13059-015-0727-9>

Moreover, as stated above Arima Genomics now offer a genome-wide Capture Hi-C for promoters on Hi-C libraries generated with MboI or DpnII plus another restriction enzyme (the identity of which is proprietary I believe) – this will bring down resolution well below 1 kb, provided sufficient sequencing depth.

4.) Comparison to Promoter Capture Hi-C:

"To illustrate this, we have pasted Figure 1 from Hua et al (Nature, 2021) below. This figure shows that PChi-C data for the alpha-globin promoter bait (Hba-a2) detects interactions with only 1 out of the 5 wellcharacterized alpha-globin enhancers in erythroid cells (panel b, enhancer locations indicated with green arrows)."

This is interesting – but as I and other reviewers have pointed out repeatedly during this reviewing process, to be fair these comparisons need to be performed comparing data sets at equal sequencing depths. I noted that Hua et al Nature 2021 sequence to a depth of 500,000 to 1,000,000 reads per view point. My understanding is that in PChi-C, this is ~100 fold lower. Have the authors taken this difference in sequencing depth into account for their comparison?

On a more general note, for balance - the direct comparisons between Capture-C and Capture Hi-C that I have seen seem to favour Capture Hi-C, for example:

<https://genomebiology.biomedcentral.com/articles/10.1186/s13059-015-0727-9>
<https://link.springer.com/article/10.1007/s00439-021-02326-8>

Having stated that a "an extensive comparison of PChi-C to Tiled-MCC data is beyond the scope of our paper.", the authors now say that: "Compared to Promoter Capture Hi-C, Tiled-MCC has significantly higher resolution and sensitivity and therefore allows for more detailed quantitative interpretation of interaction strength within a larger dynamic range. This could explain the differences in the observed effects of cohesin depletion on enhancer-promoter interactions in the Promoter Capture Hi-C and Tiled-MCC data."

I would encourage the authors to take this paragraph out. I agree that a thorough comparison between PChi-C and Tiled-MCC is beyond the scope of this manuscript. But a half-hearted attempt is even worse in my view. The statement above is biased and only describes one side of the coin. A fair comparison would involve sequencing depth, number of promoters interrogated etc. There are other explanations for the observed differences between Thiecke et al and this manuscript that go beyond the techniques used of course, most notably the cell types used in the respective studies.

My main point on Thiecke et al – which I believe I clearly made in my comments – was that the authors statement (that Thiecke et al. conclude that enhancer-promoter interactions are maintained upon cohesin depletion) was simply wrong.

REVIEWERS' COMMENTS

Reviewer #1 (Remarks to the Author):

the authors have fully addressed my points.

We thank the Reviewer for their feedback on our manuscript and are glad that all concerns have been addressed.

Reviewer #2 (Remarks to the Author):

I thank the authors for addressing my questions. Unfortunately, I have to say that I am now less convinced about the technological advances that Tiled MCC offers than I was when I initially read the manuscript for the first time. The authors admit, and this has to be recommended, that Tiled MCC has several disadvantages over Capture C or Capture Hi-C approaches, which include more a complex enrichment strategy, less efficient ligation, and fewer informative sequencing reads. The advantage is of course that Tiled-MCC does not rely on restriction sites and that the fragment size in the libraries is more uniform – but this has been known since the original Micro-C paper was published in 2015.

So the overall method development advances are limited, given that enrichment of 3C and Hi-C libraries with biotinylated oligos has been published before. And I am yet to be convinced that Tiled MCC enables any biologically meaningful discoveries that would not be possible using a Capture C or Capture Hi-C approach, especially those using 4 bp cutter restriction enzymes or combinations of restriction enzymes (see also below).

We thank the Reviewer for their feedback on our manuscript and have clarified the remaining concerns below.

Specific comments:

1.) I have some further questions about the resolution:

“However, down-sampling analyses show that 2-4-fold deeper sequencing compared to Tiled-C suffices for data visualization at 500 bp resolution (Supplementary Figures 4, 5).”

“However, since the resolution of Tiled-C is limited by the cut site distribution of restriction enzymes, it does not support data generation beyond a resolution of ~2 kb (Supplementary Figure 2).”

I am not convinced by this. ~500 bp (or even below) resolution is equivalent to what Capture-C or Capture Hi-C approaches provide when using a restriction enzyme with 4 pb recognition site (MboI or DpnII for example). The authors repeatedly use the argument that HindIII cuts every 4096 bp on average. True, but by the same logic, a restriction enzyme with a 4 bp recognition site cuts every 256 bp.

An alternative strategy is to use two restriction enzymes, which is the approach Arima Genomics are taking for their Hi-C and Capture Hi-C, further increasing resolution (see also below). I appreciated that the MCC does not rely on restriction sites and therefore offers a more uniform size representation of DNA regions. But my bet is that the average fragment size using two restriction enzymes (Arima’s approach) is very similar to that in Micro-C libraries.

The Reviewer is correct that restriction enzymes with a 4 bp recognition motif cut in theory every 256 bp. For “1D” interaction profiles from a viewpoint of interest, the resulting resolution is therefore on average 256 bp. However, given the irregular size and distribution of restriction fragments (in the mouse genome, 27% of DpnII fragments are over 500 bp and 8% are over 1 kb in size), the resolution is in practice much lower (**Figure A**). This is particularly apparent for “many vs many” or “all vs all” approaches, in which the data are represented in a “2D” contact matrix. To account for the irregular distribution of restriction sites (and consequently, data points), the bin size must be much larger to avoid having many empty bins which do not contain any data. This is clearly shown in Supplementary Figure 2, in particular in the bottom panel. When Tiled-C data (generated with DpnII-mediated digestion) are visualized at 500 bp, the enhancer-promoter interactions are difficult to identify due to the many empty bins in the matrix. We have discussed this in the previous version of our manuscript in the following paragraph:

“For enzymes with a 4 bp recognition motif (such as DpnII), the theoretical resolution limit is 256 bp; for enzymes with a 6 bp recognition motif (such as HindIII), resolution is theoretically limited to 4096 bp. However, due to the irregular distribution of the recognition motifs, bin sizes of 1-2 kb and 5-10 kb, respectively, are usually required to avoid “empty bins” for which information on chromatin interactions is missing due to the absence of restriction sites.”

Figure A: Cumulative distribution of fragment sizes generated with the *DpnII* enzyme (4 bp recognition motif) in the mouse genome.

The Reviewer is correct that use of two restriction enzymes results in higher resolution compared to use of a single restriction enzyme. However, a caveat of this approach is that it results in the formation of many extremely small restriction fragments which are not efficiently cross-linked/ligated and can also not be targeted by capture oligonucleotides. In addition, it is not possible to keep the cells intact (as in MCC) when conventional restriction enzymes are used, since the enzymes are unable to digest the chromatin sufficiently without use of strong detergents. This leads to a reduction of the signal-to-noise ratio.

Importantly, the Tiled-MCC data allow for the analysis of local micro-topologies at a resolution of 20 bp. This is not possible with approaches based on restriction enzymes, as the ligation junctions are not meaningful. We have shown this in Supplementary Figure 8 and have discussed this in the previous version of our manuscript in the following paragraph:

“The analysis of micro-topologies requires sequencing of Tiled-MCC data to a depth of at least 100-200 million reads per enriched Mb (Supplementary Figure 7). Since methods based on restriction enzyme digestion can only generate ligation junctions at restriction sites, the direct analysis of such junctions is not meaningful, as they merely reflect the distribution of restriction fragments across the genome (Supplementary Figure 8).”

2.) More on resolution:

“Because Micro-C ligation junctions are inferred, the resolution of Micro-C is limited to ~200 bp, which makes current Micro-C protocols unsuitable for the detection of very high-resolution features of chromatin structure (Supplementary Figure 9).”

“In the Tiled-MCC approach, ligation junctions are sequenced directly, because the libraries are sonicated to an average size of ~200 bp and sequenced with paired-end reads of 150 bp each.”

Is this really a true advantage of Tiled-MCC over Micro-C? Surely, Micro-C libraries could be sonicated to an average size of 200 bp and sequenced with PE 150bp, ensuring that ligation junctions are sequenced directly?

The Reviewer is correct that the Micro-C procedure could be adapted to incorporate aspects of our MCC approaches. However, this might be less straightforward than the reviewer suggests. Although it might be possible to sonicate Micro-C libraries to obtain smaller fragment sizes, there are other differences which might make it very challenging to adapt Micro-C to generate data below 200 bp resolution. First, the library complexity is more difficult to maintain in the Micro-C protocol as compared to the MCC protocol (as described in Supplementary Note 1); this will limit the resolution. Second, the MCC approach allows for use of intact cells, which markedly increases the signal-to-noise ratio. If all these modifications were made to Micro-C, it would almost completely replicate the MCC protocols.

3.) Comparison to Promoter Capture Hi-C, and again resolution:

“We do not fully agree that the Promoter Capture Hi-C (PCHi-C) approach offers sufficient resolution and sensitivity to reliably detect enhancer-promoter interactions. PCHi-C generally relies on *HindIII* digestion. Since *HindIII* has a 6 bp recognition motif, it will theoretically cut every 4096 bp.”

It is true that the majority of existing PCHi-C data sets have used *HindIII* for Hi-C library generation. However, as the authors as experts in the field well know, this is not a defining feature of PCHi-C. The restriction enzyme can easily be exchanged for a 4 bp cutter to afford higher resolution (albeit at the expense of deeper sequencing required), as demonstrated for example here:

[https://www.cell.com/cell/pdf/S0092-8674\(21\)00382-2.pdf](https://www.cell.com/cell/pdf/S0092-8674(21)00382-2.pdf)

And indeed HiCap, extremely similar to PChi-C has been established in 2015 using a 4 bp cutter: <https://genomebiology.biomedcentral.com/articles/10.1186/s13059-015-0727-9>

Moreover, as stated above Arima Genomics now offer a genome-wide Capture Hi-C for promoters on Hi-C libraries generated with MboI or DpnII plus another restriction enzyme (the identity of which is proprietary I believe) – this will bring down resolution well below 1 kb, provided sufficient sequencing depth.

The Reviewer is correct that Promoter Capture Hi-C data could be generated with restriction enzymes with a 4 bp recognition motif. As discussed in response to the first comment, this limits the resolution of “1D” interaction profiles to 256 bp and “2D” contact matrices to ~1-2 kb. Given that Tiled-MCC generates “2D” contact matrices and Promoter-Capture Hi-C generates “1D” interaction profiles it is difficult to make meaningful comparison between the approaches.

The reason we have commented on the resolution and sensitivity of Promoter Capture Hi-C data is related to the comparison of our results with the results from Thiecke et al. Since Thiecke et al use HindIII digestion, we have focused on HindIII-digested Promoter Capture Hi-C for our comparison, which is also more commonly used than DpnII/MboI-digested Promoter Capture Hi-C. As discussed in response to the next comment, we have clarified that our discussion refers to the data generated by Thiecke and colleagues and not to Promoter Capture Hi-C data in general.

4.) Comparison to Promoter Capture Hi-C:

“To illustrate this, we have pasted Figure 1 from Hua et al (Nature, 2021) below. This figure shows that PChi-C data for the alpha-globin promoter bait (Hba-a2) detects interactions with only 1 out of the 5 well characterized alpha-globin enhancers in erythroid cells (panel b, enhancer locations indicated with green arrows).”

This is interesting – but as I and other reviewers have pointed out repeatedly during this reviewing process, to be fair these comparisons need to be performed comparing data sets at equal sequencing depths. I noted that Hua et al Nature 2021 sequence to a depth of 500,000 to 1,000,000 reads per view point. My understanding is that in PChi-C, this is ~100 fold lower. Have the authors taken this difference in sequencing depth into account for their comparison?

On a more general note, for balance - the direct comparisons between Capture-C and Capture Hi-C that I have seen seem to favour Capture Hi-C, for example:

<https://genomebiology.biomedcentral.com/articles/10.1186/s13059-015-0727-9>
<https://link.springer.com/article/10.1007/s00439-021-02326-8>

Having stated that a “an extensive comparison of PChi-C to Tiled-MCC data is beyond the scope of our paper.”, the authors now say that: “Compared to Promoter Capture Hi-C, Tiled-MCC has significantly higher resolution and sensitivity and therefore allows for more detailed quantitative interpretation of interaction strength within a larger dynamic range. This could explain the differences in the observed effects of cohesin depletion on enhancer-promoter interactions in the Promoter Capture Hi-C and Tiled-MCC data.”

I would encourage the authors to take this paragraph out. I agree that a thorough comparison between PChi-C and Tiled-MCC is beyond the scope of this manuscript. But a half-hearted attempt is even worse in my view. The statement above is biased and only describes one side of the coin. A fair comparison would involve sequencing depth, number of promoters interrogated etc. There are other explanations for the observed differences between Thiecke et al and this manuscript that go beyond the techniques used of course, most notably the cell types used in the respective studies.

My main point on Thiecke et al – which I believe I clearly made in my comments – was that the authors statement (that Thiecke et al. conclude that enhancer-promoter interactions are maintained upon cohesin depletion) was simply wrong.

We are happy that the Reviewer agrees that a detailed comparison between these approaches is beyond the scope of our paper. We will therefore restrict our discussion to the data generated by Thiecke et al. Since these data have been generated using HindIII digestion, the resolution is much lower compared to the Tiled-MCC data in our study. We have adapted our statement as follows:

“Compared to the Promoter Capture Hi-C data generated *in this study*, Tiled-MCC has significantly higher resolution and sensitivity and therefore allows for more detailed quantitative interpretation of interaction strength within a larger dynamic range.”